# GeneDisco: A Benchmark for Experimental Design in Drug Discovery

**Arash Mehrjou**[1]**, Ashkan Soleymani**[2]**, Andrew Jesson**[3]**, Pascal Notin**[3]**,
Yarin Gal**[3]**, Stefan Bauer**[1]**, Patrick Schwab**[1]
[1]GlaxoSmithKline, Artificial Intelligence & Machine Learning
[2] MIT, [3] Department of Computer Science, University of Oxford

## Abstract

In vitro cellular experimentation with genetic interventions, using for example CRISPR technologies, is an essential step in early-stage drug discovery and target validation that serves to assess initial hypotheses about causal associations between biological mechanisms and disease pathologies. With billions of potential hypotheses to test, the experimental design space for in vitro genetic experiments is extremely vast, and the available experimental capacity - even at the largest research institutions in the world - pales in relation to the size of this biological hypothesis space. Machine learning methods, such as active and reinforcement learning, could aid in optimally exploring the vast biological space by integrating prior knowledge from various information sources as well as extrapolating to yet unexplored areas of the experimental design space based on available data. However, there exist no standardised benchmarks and data sets for this challenging task and little research has been conducted in this area to date. Here, we introduce GeneDisco, a benchmark suite for evaluating active learning algorithms for experimental design in drug discovery. GeneDisco contains a curated set of multiple publicly available experimental data sets as well as open-source implementations of state-of-the-art active learning policies for experimental design and exploration.

## 1 Introduction

The discovery and development of new therapeutics is one of the most challenging human endeavours with success rates of around 5% (Hay et al., 2014; Wong et al., 2019), timelines that span on average over a decade (Dickson & Gagnon, 2009; 2004), and monetary costs exceeding two billion United States (US) dollars (DiMasi et al., 2016; Berdigaliyev & Aljofan, 2020). The successful discovery of drugs at an accelerated pace is critical to satisfy current unmet medical needs (Rawlins, 2004; Ringel et al., 2020), and, with thousands of potential treatments currently in development (informa PLC, 2018), increasing the probability of success of new medicines by establishing causal links between drug targets and diseases (Nelson et al., 2015) could introduce an additional hundreds of new and potentially life-changing therapeutic options for patients every year.

However, given the current estimate of around 20 000 protein-coding genes (Pertea et al., 2018), a continuum of potentially thousands of cell types and states under a multitude of environmental conditions (Trapnell, 2015; MacLean et al., 2018; Worzfeld et al., 2017), and tens of thousands of cellular measurements that could be taken (Hasin et al., 2017; Chappell et al., 2018), the combinatorial space of biological exploration spans hundreds of billions of potential experimental configurations, and vastly exceeds the experimental capacity of even the world's largest research institutes. Machine learning methods, such as active and reinforcement learning, could potentially aid in optimally exploring the space of genetic interventions by prioritising experiments that are more likely to yield mechanistic insights of therapeutic relevance (Figure 1), but, given the lack of openly accessible curated experimental benchmarks, there does not yet exist to date a concerted effort to leverage the machine learning community for advancing research in this important domain.

To bridge the gap between machine learning researchers versed in causal inference and the challenging task of biological exploration, we introduce GeneDisco, an open benchmark suite for evaluating batch active learning algorithms for experimental design in drug discovery. GeneDisco consists of

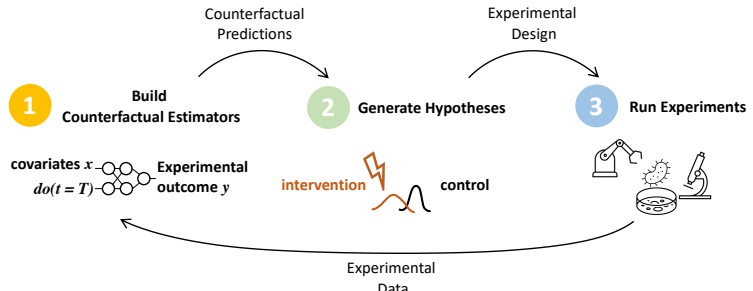

Figure 1: In the setting considered in this work, counterfactual estimators of experimental outcomes (step 1, left) are used to propose experimental hypotheses (step 2, center) for validation in in vitro experiments with genetic interventions (step 3, right), such as CRISPR knockouts, in order to discover potential causal associations between biological entities that could be relevant for the development of novel therapeutics. The trained counterfactual estimators can be used to direct the experimental search towards the space of biological interest, and thus more efficiently explore the vast space of genetic interventions. After every cycle, experimental data are generated that could lead to mechanistic scientific discoveries forming the basis for new therapeutics development, and guide subsequent experiment cycles with enhanced counterfactual estimators.

several curated datasets, tasks and associated performance metrics, open implementations of state-of-the-art active learning algorithms for experimental design, and an accessible open-source code base for evaluating and comparing new batch active learning methods for biological discovery.

Concretely, the contributions presented in this work are as follows:

- We introduce GeneDisco, an open benchmark suite for batch active learning for drug discovery that provides curated datasets, tasks, performance evaluation and open source implementations of state-of-the-art algorithms for experimental exploration.
- We perform an extensive experimental baseline evaluation that establishes the relative performance of existing state-of-the-art methods on all the developed benchmark settings using a total of more than 20 000 central processing unit (CPU) hours of compute time.
- We survey and analyse the current state-of-the-art of active learning for biological exploration in the context of the generated experimental results, and present avenues of heightened potential for future research based on the developed benchmark.

## 2   RELATED WORK

**Background.** Drug discovery is a challenging endeavour with (i) historically low probabilities of successful development into clinical-stage therapeutics (Hay et al., 2014; Wong et al., 2019), and, for many decades until recently (Ringel et al., 2020), (ii) declining industry productivity commonly referred to as "Eroom's law" (Scannell et al., 2012). Seminal studies by Nelson et al. (2015) and King et al. (2019) respectively first reported and later independently confirmed that the probability of clinical success of novel therapeutics increases up to three-fold if a medicine's molecular target is substantiated by high-confidence causal evidence from genome-wide association studies (GWAS) Visscher et al. (2017). With the advent of molecular technologies for genetic perturbation, such as Clustered Regularly Interspaced Short Palindromic Repeats (CRISPR) (Jehuda et al., 2018), there now exist molecular tools for establishing causal evidence supporting the putative mechanism of a potential therapeutic hypothesis by means of in vitro experimentation beyond GWAS early on during the target identification and target validation stages of drug discovery (Rubin et al., 2019; Itokawa et al., 2016; Harrer et al., 2019; Vamathevan et al., 2019). Among other applications (Chen et al., 2018; Ekins et al., 2019; Mak & Pichika, 2019), machine learning methods, such as active and reinforcement learning, could potentially aid in discovering the molecular targets with the highest therapeutic potential faster.

**Machine Learning for Drug Discovery.** There exist a number of studies that proposed, advanced or evaluated the use of machine learning algorithms for drug discovery: Costa et al. (2010) used decision tree meta-classifier to identify genes associated with morbidity using protein-protein, metabolic, transcriptional, and sub-cellular interactions as input. Jeon et al. (2014) used a support vector machine (SVM) on protein expressions to predict target or non-target for breast pancreatic or ovarian cancers, and Ament et al. (2018) used least absolute shrinkage and selection operator (LASSO)

regularised linear regression on transcription factor binding sites and transcriptome profiling to predict transcriptional changes associated with Huntington's disease. In the domain of human muscle ageing, Mamoshina et al. (2018) used a SVM on deep features extracted from gene expression signatures in tissue samples from young and old subjects to discover molecular targets with putative involvement in muscle ageing. More recently, Stokes et al. (2020) utilised deep learning to discover Halicin as a repurposed molecule with antibiotic activity in mice.

**Reinforcement and Active Learning.** The use of deep reinforcement learning for de novo molecular design has been extensively studied (Olivecrona et al., 2017; Popova et al., 2018; Putin et al., 2018; Lim et al., 2018; Blaschke et al., 2020; Gottipati et al., 2020; Horwood & Noutahi, 2020). Active learning for de novo molecular design has seen less attention (Dixit et al., 2016; Green et al., 2020), however, active learning for causal inference has seen increasing application toward causal-effect estimation (Deng et al., 2011; Schwab et al., 2018; Sundin et al., 2019; Schwab et al., 2020; Bhattacharyya et al., 2020; Parbhoo et al., 2021; Qin et al., 2021; Jesson et al., 2021; Chau et al., 2021), and causal graph discovery (Ke et al., 2019; Tong & Koller, 2001; Murphy, 2001; Hauser & Bühlmann, 2014; Ghassami et al., 2018; Ness et al., 2017; Agrawal et al., 2019; Lorch et al., 2021; Annadani et al., 2021; Scherrer et al., 2021).

**Benchmarks.** Benchmark datasets play an important role in developing machine learning methodologies. Examples include ImageNet (Deng et al., 2009) or MSCOCO (Lin et al., 2014) for computer vision, as well as cart-pole (Barto et al., 1983) or reinforcement learning (Ahmed et al., 2020). Validation of active learning for causal inference methods depends largely on synthetic data experiments due to the difficulty or impossibility of obtaining real world counterfactual outcomes. For causal-effect active learning, real world data with synthetic outcomes such as IHDP (Hill, 2011) or ACIC2016 Dorie et al. (2019) are used. For active causal discovery, *in silico* data such as DREAM4 (Prill et al., 2011) or the gene regulatory networks proposed by Marbach et al. (2009) are used. Non synthetic data has been limited to protein signalling networks (Sachs et al., 2005) thus far. Several benchmarks focus on some aspects of drug discovery, for instance Brown et al. (2019) focusing on de novo molecular screening. However, to our knowledge, no other benchmark addressing the target identification part of the drug discovery pipeline, which is what our work focuses on.

In contrast to existing works, we develop an open benchmark to evaluate the use of machine learning for efficient experimental exploration in an iterative batch active learning setting. To the best of our knowledge, this is the first study (i) to introduce a curated open benchmark for the challenging task of biological discovery, and (ii) to comprehensively survey and evaluate state-of-the-art active learning algorithms in this setting.

## 3 METHODOLOGY

**Problem Setting.** We consider the setting in which we are given a dataset consisting of covariates $X \in \mathbb{R}^p$ with input feature dimensionality $p \in \mathbb{N}$ and treatment descriptors $T \in \mathbb{R}^q$ with treatment descriptor dimensionality $q \in \mathbb{N}^+$ that indicate similarity between interventions. Our aim is to estimate the expectation of the conditional distribution of an unseen counterfactual outcome $Y_t \in \mathbb{R}$ given observed covariates $X = x$ and intervention $do(T = t)$, $\hat{y}_t = \widehat{g}(X = x, do(T = t)) \approx \mathbb{E}[Y \mid X = x, do(T = t)]$. This setting corresponds to the Rubin-Neyman potential outcomes framework (Rubin, 2005) adapted to the context of genetic interventions with a larger number of parametric interventions. In the context of a biological experiment with genetic interventions, $y_t$ is the experimental outcome relative to a non-interventional control (e.g., change in pro-inflammatory effect) that is measured upon perturbation of the cellular system with intervention $t$, $x$ is a descriptor of the properties of the model system and/or cell donor (e.g., the immuno-phenotype of the cell donor), and $t$ is a descriptor of the genetic intervention (e.g., a CRISPR knockout on gene STAT1) that indicates similarity to other genetic interventions that could potentially be applied. In general, certain ranges of $y_t$ may be preferable for further pursuit of an intervention $T = t$ that inhibits a given molecular target - often, but not necessarily always, larger absolute values that move the experimental result more are of higher mechanistic interest. We note that the use of an empty covariate vector $X = x_0$ with $p = 0$ is permissible if all experiments are performed in the same model system with the same donor properties. In in vitro experimentation, the set of all possible genetic interventions $\mathcal{D}_{\text{pool}} = \{t_i\}_{i=1}^{n_{\text{pool}}}$ is typically known a-priori and of finite size (e.g., knockout interventions on all 20 000 protein-coding genes).

**Batch Active Learning.** In the considered setting, reading out additional values for yet unexplored interventions $t$ requires a lab experiment and can be costly and time-consuming. Lab experiments are typically conducted in parallelised manner, for example by performing a batch of multiple interventions at once in an experimental plate. Our overall aim is to leverage the counterfactual estimator $\widehat{g}$ trained on the available dataset to simulate future experiments with the aim of maximising an objective function $\mathcal{L}$ in the next iteration with as few interventions as possible. For the purpose of this benchmark, we consider the counterfactual mean squared error (MSE) of $\widehat{g}$ in predicting experimentally observed true outcomes $y_t$ from predicted outcomes $\hat{y}_t$ as the optimisation objective $\mathcal{L}_{\mathrm{MSE}} = \mathrm{MSE}(y_t, \hat{y}_t)$. Depending on context, other objective functions, such as for example maximising the number of discovered molecular targets with certain properties of interest (e.g., druggability (Keller et al., 2006)) may also be sensible in the context of biological exploration. At every time point, a new counterfactual estimator $\widehat{g}$ is trained with the entire available experimental dataset, and used to propose the batch of $b$ interventions to be explored in the next iteration with the batch size $b \in \mathbb{N}^+$. When using $\mathcal{L}_{\mathrm{MSE}}$, this setting corresponds to batch active learning with the optimisation objective of maximally improving the counterfactual estimator $\widehat{g}$.

**Acquisition Function.** An acquisition function $\mathcal{D}^k = \alpha(g(t), \mathcal{D}^k_{\mathrm{avail}})$ takes the model and the set of all available interventions $\mathcal{D}^k_{\mathrm{avail}}$ in cycle $k$ as input and outputs the set of interventions $\mathcal{D}^k$ that are most informative after the $k$th experimental cycle with cycle index $k \in [K] = [0 \, .. \, K]$ where $K \in \mathbb{N}^+$ is the maximum number of cycles that can be run. Formally speaking, the acquisition function $\alpha : \mathcal{P}(\mathcal{D}_{\mathrm{avail}}) \times \mathcal{G} \to \mathcal{P}(\mathcal{D}_{\mathrm{avail}})$ takes a subset of all possible interventions that have not been tried so far ($\mathcal{D}_{\mathrm{avail}}$), together with the trained model ($\widehat{g}$) derived from the cumulative data collected over the previous cycles, and outputs a subset of the available interventions $\mathcal{D}^k$ that are likely to be most useful under $\mathcal{L}$ to obtain $\widehat{g} \in \mathcal{G}$ as a better estimate of $\mathbb{E}[Y \mid X = x, do(T = t)]$ with $\mathcal{G}$ being the space of the models which can be, e.g. the space of models and (hyper-)parameters.

## 4 DATASETS, METRICS & BASELINES

The GeneDisco benchmark curates and standardizes two types of datasets: three standardized feature sets describing interventions $t$ (inputs to counterfactual estimators; Section 4.1), and four different in vitro genome-wide CRISPR experimental assays (predicted counterfactual outcomes; Section 4.2), each measuring a specific outcome $y_t$ following possible interventions $T$. We perform an extensive evaluation across these datasets, leveraging two different model types (Section 4.3) and nine different acquisition functions (Section 4.4). Since all curated assay datasets contained only outcomes for only one model system, we used the empty covariate set $X = x_0$ for all evaluated benchmark configurations. The metrics used to evaluate the various experimental conditions (acquisition functions and model types) include model performance (Figure 2) and the ratio of discovered interesting hits (Figure 3) as a function of number of samples queried.

### 4.1 TREATMENT DESCRIPTORS

The treatment descriptors $T$ characterize a genetic intervention and generally should correspond to data sources that are informative as to a genes' functional similarity - i.e. defining which genes if acted upon, would potentially respond similarly to perturbation. The treatment descriptor $T$ is the input to the model described in Section 3 for a fixed vector of covariates $x$. Any dataset considered for use as a treatment descriptor must be available for all potentially available interventions $\mathcal{D}_{\mathrm{pool}}$ in the considered experimental setting. In GeneDisco, we provide three standardised gene descriptor sets for genetic interventions, and furthermore enable users to provide custom treatment descriptors via a standardised programming interface:

**Achilles.** The Achilles project generated dependency scores across cancer cell lines by assaying 808 cell lines covering a broad range of tissue types and cancer types (Dempster et al., 2019). The genetic intervention effects are based on interventional CRISPR screens performed across the included cell lines. When using the Achilles treatment descriptors, each genetic intervention is summarized using a gene representation $T$ with $q = 808$ corresponding to the dependency scores measured in each cell line. In Achilles, after processing and normalisation (see Dempster et al. (2019)), the final dependency scores provided are such that the median negative control (non-essential) gene effect for each cell line is 0, and the median positive control (essential) gene effect for each cell line is -1. The rationale for using treatment descriptors based on the Achilles dataset is that genetic

effects measured across the various tissues and cancer types in the 808 cell line assays included in (Dempster et al., 2019) could serve as a similarity vector in functional space that may extrapolate to other biological contexts due to its breadth.

**Search Tool for the Retrieval of Interacting Genes/Proteins (STRING) Network Embeddings.** The STRING (Szklarczyk et al., 2021) database collates known and predicted protein-protein interactions (PPIs) for both physical as well as for functional associations. In order to derive a vector representation suitable to serve as a genetic intervention descriptor $T$, we utilised the network embeddings of the PPIs contained in STRING as provided by Cho et al. (2016; 2015) with dimensionality $q = 799$. PPI network embeddings could be an informative descriptor of functional gene similarity since proteins that functionally interact with the same network partners may serve similar biological functions (Vazquez et al., 2003; Saha et al., 2014).

**Cancer Cell Line Encyclopedia (CCLE).** The CCLE (Nusinow et al., 2020) project collected quantitative proteomics data from thousands of proteins by mass spectrometry across 375 diverse cancer cell lines. The generated protein quantification profiles with dimensionality $q = 420$ could indicate similarity of genetic interventions since similar expression profiles across a broad range of biological contexts may indicate functional similarity.

**Custom Treatment Descriptors.** Additional, user-defined treatment descriptors can be evaluated in GeneDisco by implementing the standardised dataset interface provided within.

## 4.2 ASSAYS

As ground-truth interventional outcome datasets, we leverage various genome-wide CRISPR screens, primarily from the domain of immunology, that evaluated the causal effect of intervening on a large number of genes in cellular model systems in order to identify the genetic perturbations that induce a desired phenotype of interest. In connection with the problem formulation in Section 3, the CRISPR screens are the random variables whose realized values are the outcome of the interventional experiments. In the following, we add more details about four CRISPR screens.

### 4.2.1 REGULATION OF HUMAN T CELLS PROLIFERATION

**Experimental setting.** This assay is based on Shifrut et al. (2018). After isolating CD8$^+$ T cells from human donors, Shifrut et al. (2018) performed a genome-wide loss-of-function screen to identify genes that impact the proliferation of T cells following stimulation with T cell receptors.

**Measurement.** The measured outcome is the proliferation of T cells in response to T cell receptor stimulation. Cells were labeled before stimulation with CFSE (a fluorescent cell staining dye). Proliferation of cells is measured 4 days following stimulation by FACS sorting (a flow cytometry technique to sort cells based on their fluorescence).

**Importance.** Human T cells play a central role in immunity and cancer immunotherapies. The identification of genes driving T cell proliferation could serve as the basis for new preclinical drug candidates or adoptive T cell therapies that help eliminate cancerous tumors.

### 4.2.2 INTERLEUKIN-2 PRODUCTION IN PRIMARY HUMAN T CELLS

**Experimental setting.** This dataset is based on a genome-wide CRISPR interference (CRISPRi) screen in primary human T cells to uncover the genes regulating the production of Interleukin-2 (IL-2). CRISPRi screens test for loss-of-function genes by reducing their expression levels. IL-2 is a cytokine produced by CD4$^+$ T cells and is a major driver of T cell expansion during adaptive immune responses. Assays were performed on primary T cells from 2 different donors. The detailed experimental protocol is described in Schmidt et al. (2021).

**Measurement.** Log fold change (high/low sorting bins) in IL-2 normalized read counts (averaged across the two biological replicates for robustness). Sorting was done via flow cytometry after intracellular cytokine staining.

**Importance.** IL-2 is central to several immunotherapies against cancer and autoimmunity.

### 4.2.3 Interferon-$\gamma$ production in primary human T cells

**Experimental setting.** This data is also based on Schmidt et al. (2021), except that this experiment was performed to understand genes driving production of Interferon-$\gamma$ (IFN-$\gamma$). IFN-$\gamma$ is a cytokine produced by CD4$^+$ and CD8$^+$ T cells that induces additional T cells.

**Measurement.** Log fold change (high/low sorting bins) in IFN-$\gamma$ normalized read counts (averaged across the two biological replicates for robustness).

**Importance.** IFN-$\gamma$ is critical to cancerous tumor killing and resistance to IFN-$\gamma$ is one escape mechanism for malignant cells.

### 4.2.4 Vulnerability of Leukemia cells to NK cells

**Experimental setting.** This genome-wide CRISPR screen was performed in the K562 cell line to identify genes regulating the sensitivity of leukemia cells to cytotoxic activity of primary human NK cells. Detailed protocol is described in Zhuang et al. (2019).

**Measurement.** Log fold counts of gRNAs in surviving K562 cells (after exposition to NK cells) compared to control (no exposition to NK cells). Gene scores are normalized fold changes for all gRNAs targeting this gene.

**Importance.** Better understanding and control over the genes that drive the vulnerability of leukemia cells to NK cells will help improve anti-cancer treatment efficacy for leukemia patients, for example by preventing relapse during hematopoeitic stem cell transplantation.

### 4.3 Models

Parametric or non-parametric models can be used to model the conditional expected outcomes, $\mathbb{E}[Y \mid X = x, do(T = t)]$. Parametric models assume that the outcome $Y$ has density $f(y \mid t, \omega)$ conditioned on the intervention $t$ and the parameters of the model $\omega$ (we drop $x_0$ for compactness). A common assumption for continuous outcomes is a Gaussian distribution with density $f(y \mid t, \omega) = \mathcal{N}(y \mid \widehat{g}(t; \omega), \sigma^2)$, which assumes that $y$ is a deterministic function of $\widehat{g}(t; \omega)$ with additive Gaussian noise scaled by $\sigma^2$. Bayesian methods treat the model parameters $\omega$ as instances of the random variable $\Omega \in \mathcal{W}$ and aim to model the posterior density of the parameters given the data, $f(\omega \mid \mathcal{D})$. For high-dimensional, large-sample data, such as we explore here, a variational approximation to the posterior is often used, $q(\omega \mid \mathcal{D})$ (MacKay, 1992; Hinton & Van Camp, 1993; Barber & Bishop, 1998; Gal & Ghahramani, 2016). In this work we use Bayesian Neural Networks (BNNs) to approximate the posterior over model parameters. A BNN gives $\widehat{g}^{k-1}(t) = \frac{1}{m} \sum_{j=1}^{m} \widehat{g}(t; \omega_j^{k-1})$, where $\widehat{g}(t; \omega_j^{k-1})$ is a unique functional estimator of $\mathbb{E}[Y \mid X = x, do(T = t)]$ induced by $\omega_j^{k-1} \sim q(\omega \mid \mathcal{D}_{\text{cum}}^{k-1})$: a sample from the approximate posterior over parameters given the cumulative data at acquisition step $k - 1$. We also use non-parametric, non-Bayesian, Random Forest Regression (Breiman, 2001). A Random Forest gives $\widehat{g}^{k-1}(t) = \frac{1}{m} \sum_{j=1}^{m} \widehat{g}_j^{k-1}(t)$, where $\widehat{g}_j^{k-1}(t)$ is a unique functional estimator of $\mathbb{E}[Y \mid X = x, do(T = t)]$ indexed by the $j$th sample in the ensemble of $m$ trees trained on $\mathcal{D}_{\text{cum}}^{k-1}$.

We emphasize two main assumptions behind the above modeling choices for the problem formulation of Section 3: **(1)** A causal link between from the gene intervention to the phenotype screen exists. **(2)** Gaussian noise is a reasonable choice for the output noise as the interventional outcome is a continuous variable. We leave it to future work to explore heterogeneity in output noise and other likelihood functions.

In the following, we will define our acquisition functions in terms of parametric models, but the definitions are easily adapted for non-parametric models as well.

### 4.4 Acquisition Functions

We included a diverse set of the existing active learning methods to serve as an informative set of baselines for comparison and intuition activation for future improvements. Most of the Different acquisition function methods can be categorized in the following way:

- *Random*: **Random** acquisition function is an integral part of whatever baseline set in active learning's performance criteria.

- *Uncertainty-based acquisition functions*: These categories are popular because of their easy-to-develop and low-computational properties while having great performance. Uncertainty-based functions usually are employed in the shallow models that some sense of intrinsic uncertainty exists for them, e.g., random forests. Here, we adapted some of them to our notion of uncertainty (based on our regression task). These functions include `topuncertain` and **Margin Sample**.

- *Deep Bayesian Active Learning acquisition functions*: One way to bridge between classical uncertainty-based algorithms and deep active learning methods is to leverage deep bayesian active learning. Bayesian Active Learning by Disagreement (**BALD**) calculates mutual information between samples and model parameters as a definition for uncertainty.

- *Density-based acquisition functions*: These functions try to select a sample that is the best representative of the whole dataset in terms of diversity. Based on different repesentativee power measures various methods exist that here **Coreset** and `kmeansdata` are covered.

- *Adversarial learning attacks*: By choosing adversarial samples as nominated samples for active learning, one may use most of the adversarial learning algorithms as acquisition functions. We implemented **AdvBIM** as a compelling example of this category.

- *Hybrid acquisition functions*: These methods take into account both sample diversity and also prediction uncertainty of the model. In other words, they increase the diversity of data samples according to model properties. **BADGE** considers the gradient of the loss function w.r.t weights of the final layer while `kmeansembed` looks at the data from their embeddings in the last layer of the deep learning model.

Each one of these acquisition functions (Random, BADGE, BALD (`topuncertain` and `softuncertain`)), Coreset, Margin Sample (`Margin`), Adversarial Basic Iterative Method (`AdvBIM`), $k$-means Sampling (`kmeansdata` and `kmeansembed`)) included in the benchmark are described in detail in Appendix B.

## 5 EXPERIMENTAL EVALUATION

**Setup.** In order to assess current state-of-the-art methods on the GeneDisco benchmark, we perform an extensive baseline evaluation of 9 acquisition functions, 6 acquisition batch sizes and 4 experimental assays using in excess of 20 000 CPU hours of compute time. Due to the space limit, we include the results for 3 batch sizes in the main text and present the results for all batch sizes in the appendix. The employed counterfactual estimator $\hat{g}$ is a multi-layer perceptron (MLP) that has one hidden layer with ReLU activation and a linear output layer. The size of the hidden layer is determined at each active learning cycle by k-fold cross validation against 20% of the acquired batch. At each cycle, the model is trained for at most 100 epochs but early stopping may interrupt training earlier if the validation error does not decrease. Each experiment is repeated with 5 random seeds to assess experimental variance. To choose the number of active learning cycles, we use the following strategy: the number of cycles are bounded to 40 for the acquisition batches of sizes 16, 32 and 64 due to the computational limits. For larger batch sizes, the number of cycles are reduced proportionally so that the same total number of data points are acquired throughout the cycles. At each cycle, the model is trained from scratch using the data collected up to that cycle, i.e. a trained model is not transferred to the future cycles. The test data is a random 20% subset of the whole data that is left aside before the active learning process initiates, and is kept fixed across all experimental settings (i.e., for different datasets and different batch sizes) to enable a consistent comparison of the various acquisition functions, counterfactual estimator and treatment descriptor configurations.

**Results.** The model performance based on the STRING treatment descriptors for different acquisition functions, acquisition batch sizes and datasets are presented in Figure 2. The same metrics in the Achilles and CCLE treatment descriptors are provided in Appendix C. To showcase the effect of the model class, we additionally repeat the experiments using a random forest model as an uncertainty aware ensemble model with a reduced set of acquisition functions that are compatible with non-differentiable models (Appendix C). The random forest model is also used in another set of experiments with other datasets (Sanchez et al., 2021; Zhu et al., 2021) which are fully postponed to

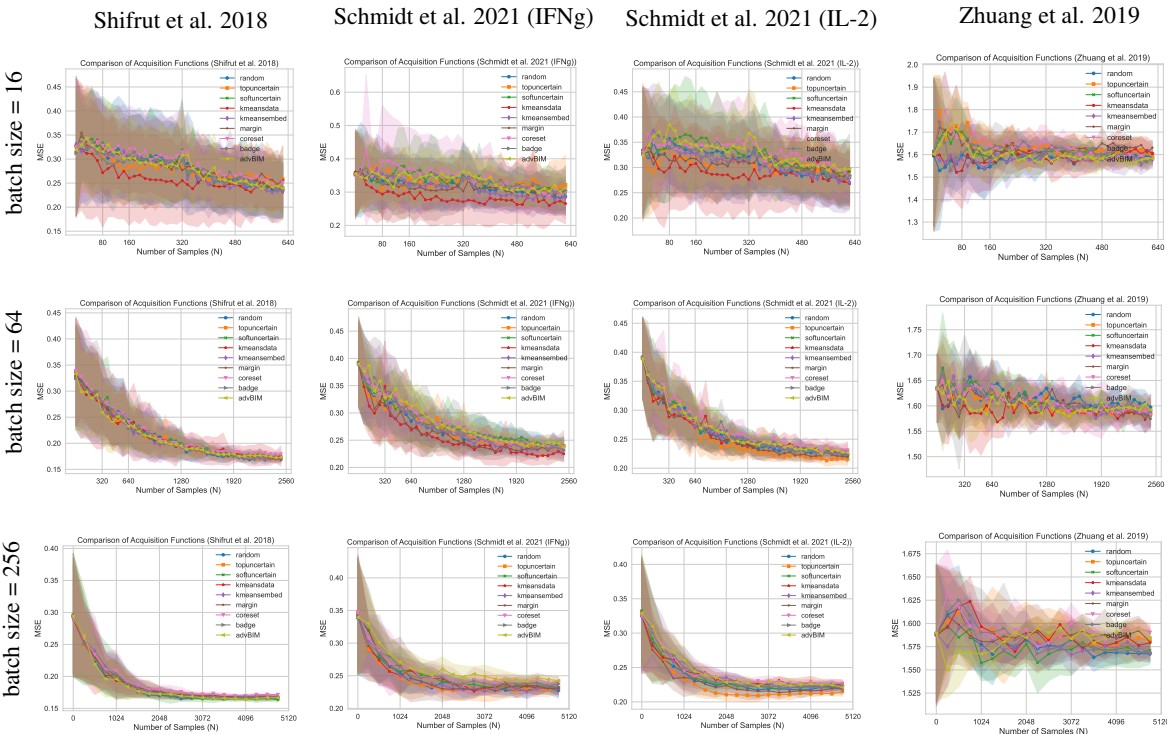

Figure 2: The evaluation of the model trained with STRING treatment descriptors at each active learning cycle for 4 datasets and 3 acquisition batch sizes. In each plot, the x-axis is the active learning cycles multiplied by the acquisition bath size that gives the total number of data points collected so far. The y-axis is the test MSE error evaluated on the test data.

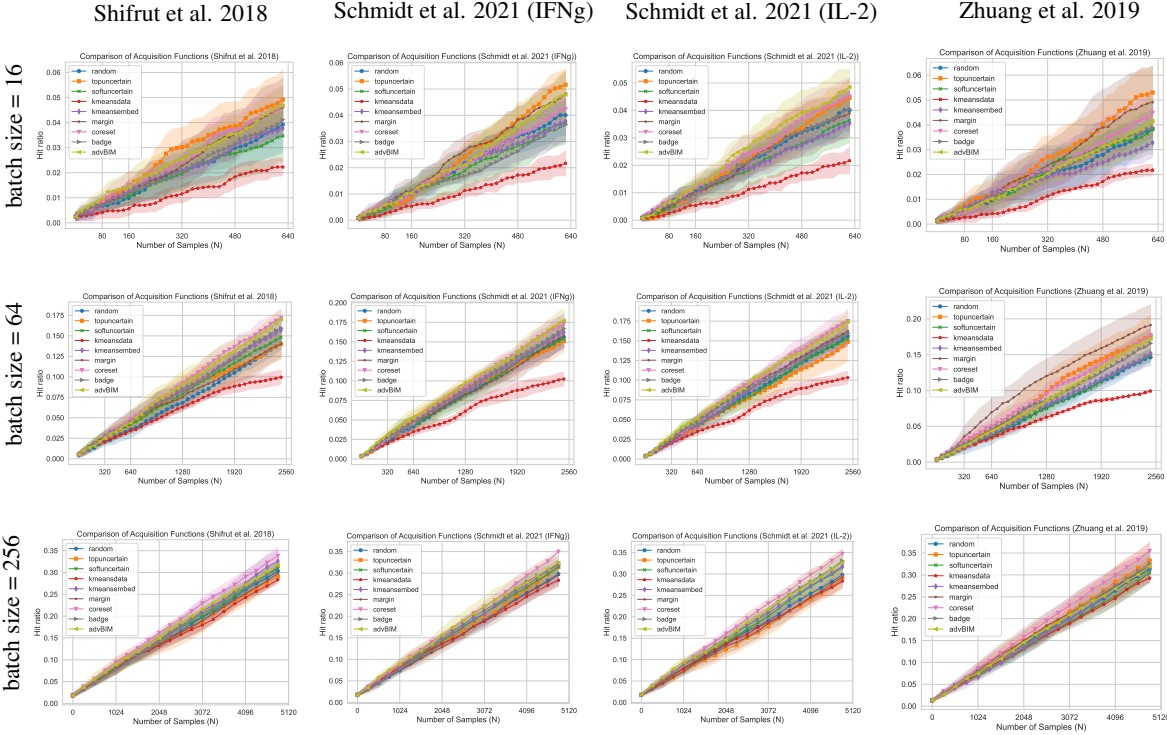

Figure 3: The hit ratio of different acquisition for BNN model, different target datasets, and different acquisition batch sizes. We use STRING treatment descriptors here. The x-axis shows the number of data points collected so far during the active learning cycles. The y-axis shows the ratio of the set of interesting genes that have been found by the acquisition function up until each cycle.

the appendix due to the space constraint. To investigate the types of genes chosen by different acquisition functions, we defined a subset of potentially interesting genes as the top $5\%$ with the largest absolute target value. These are the genes that could potentially be of therapeutic value due to their outsized causal influence on the phenotype of interest. The hit ratio out of the set of interesting genes chosen by different acquisition functions are presented in Figure 3 for the STRING treatment descriptors, and in Appendix C for Achilles and CCLE. Benchmark results of interest include that model-independent acquisition methods using diversity heuristics (random, kmeansdata) perform relatively better in terms of model improvement than acquisition functions based on model uncertainty (e.g., topuncertain, softuncertain) when using lower batch acquisition sizes than in regimes with larger batch acquisition sizes potentially due to diversity being inherently higher in larger batch acquisition regimes due to the larger set of included interventions in an intervention space with a limited amount of similar interventions. Notably, while diversity-focused, model-independent acquisition functions, such as random and kmeansdata, perform well in terms of model performance, they underperform in terms of interesting hits discovered as a function of acquired interventional samples (Figure 3). Based on these results, there appears to be a trade-off between model improvement and hit discovery in experimental exploration with counterfactual estimators that may warrant research into approaches to manage this trade-off to maximize long-term discovery rates.

## 6  DISCUSSION AND CONCLUSION

The ranking of acquisition functions in GeneDisco depends on several confounding factors, such as the choice of evaluation metric to compare different approaches, the characteristics of the dataset of interest, and the choice of the model class and its hyperparameters. An extrapolation of results obtained in GeneDisco to new settings may not be possible under significantly different experimental conditions. There is a subtle interplay between the predictive strength of the model and the acquisition function used to select the next set of interventions, as certain acquisition functions are more sensitive to the ability of the model to estimate its own epistemic uncertainty. An important message learned from Figures 2 and 3 is the observation that in active learning, the best acquisition function is the one that gives a more accurate model with the same budget of experiments. However, in drug discovery, learning an accurate predictive model has secondary importance. The primary objective is to find the optimal set of targets for a potential medicine. This objective may be unaligned or even in the opposite direction of training the predictive model. For example, even though the Kmeans acquisition function that chooses the centroids in the data domain has a decent performance in the active learning task (finding the most informative labeled dataset for training the predictive model), it does not perform well in choosing the most interesting targets (those that moves the phenotype maximally.) A potential reason could be the intuition that the interesting targets may be close to each other in terms of the information content they provide to the predictive model. The acquisition function then considers them as redundant and does not choose in the next iteration. Hence, AL algorithms must be used with care in drug discovery. We postpone further study of a more suitable objective function for the acquisition function to future work.

From a practical standpoint, GeneDisco assumes the availability of a labeled set that is sufficiently representative to train and validate the different models required by the successive active learning cycles. However, model validation might be challenging when this set is small (e.g., during the early active learning cycles) or when the labelling process is noisy. Label noise is common in interventional biological experiments, such as the ones considered in GeneDisco. Experimental noise introduces additional trade-offs for consideration not considered in GeneDisco, such as choosing the optimal budget allocation between performing experiment replicates (technical and biological) to mitigate label noise or collecting more data points via additional active learning cycles.

GeneDisco addresses the current lack of standardised benchmarks for developing batch active learning methods for experimental design in drug discovery. GeneDisco consists of several curated datasets for experimental outcomes and genetic interventions, provides open source implementations of state-of-the-art acquisition functions for batch active learning, and includes a thorough assessment of these methods across a wide range of hyperparameter settings. We aim to attract the broader active learning community with an interest in causal inference by providing a robust and user-friendly benchmark that diversifies the benchmark repertoire over standard vision datasets. New models and acquisition functions for batch active learning in experimental design are of critical importance to realise the potential of machine learning for improving drug discovery. As future research, we aim to expand GeneDisco to enable multi-modal learning and support simultaneous optimization across multiple output phenotypes of interest.

ACKNOWLEDGEMENT

PS, AM and SB are employees of GlaxoSmithKline (GSK) and PS is also a shareholder of GSK. PN is supported by GSK and the UK Engineering and Physical Sciences Research Council (EPSRC ICASE award no. 18000077).

REPRODUCIBILITY STATEMENT

This work introduces a new curated and standardized benchmark, GeneDisco, for batch active learning for drug discovery. The benchmark includes four publicly available datasets, which have previously been published in a peer review process. Using a total of more than 20,000 central processing unit (CPU) hours of compute time, we perform an extensive evaluation of state-of-the-art acquisition functions for batch active learning on the GeneDisco benchmark, across a wide range of hyperparameters. To the best of our knowledge, this is the first comprehensive survey and evaluation of active learning algorithms on real-world interventional genetic experiment data. Similar to developments in other fields e.g. for the learning of disentangled representations (Locatello et al., 2019) or generative adversarial networks (Lucic et al., 2017), we hope that our large scale experiments across a diverse set of real-world datasets provide an evidence basis to better understand the settings in which different active learning approaches work or do not work for drug discovery applications.

All used models and acquisition functions are described in detail and referenced in Section 4.4 and Section 4.3. For all introduced datasets, we include a detailed description and the details on the train, test and validation splits at the beginning of Section 5.

For all experimental results we report the range of hyper-parameters considered and the methods of selecting hyper-parameters as well as the exact number of training and evaluation runs (Appendix C). We additionally provide error bars over multiple random seeds and the code was executed on a cloud cluster with Intel CPUs. We provide detailed results for all investigated settings in the appendix (C).

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

## A    NOTATIONS

Here is the list of notations used in this manuscript.

- $\mathcal{D}_{\text{pool}}$: The pool of unlabeled data.
- $\mathcal{D}_{\text{acq}}^k$: Acquired data at $k$-th AL cycle.
- $\mathcal{D}_{\text{train}}^k$: Cumulative training data after $k$-th cycle of AL.
- $\mathcal{D}_{\text{val}}$: Validation data.
- $\mathcal{D}_{\text{test}}$: Held-out test data.
- $b$: Acquisition batch size.
- $K$: Total number of AL cycles.
- $k = [1, 2, \ldots, K]$: Index of the AL cycle.
- $[K] = [1, 2, \ldots, K]$.
- $e \in \mathcal{E}$: Inherent noise (aleatoric uncertainty).
- $T \in \mathcal{T}$: Treatment variable.
- $\mathbb{E}[Y \mid X = x, do(T = t)]$: The conditional expected outcomes.
- $\hat{g}(t; \omega)$: The model parameterised by $\omega \in \Omega$ to estimate $\mathbb{E}[Y \mid X = x, do(T = t)]$.
- $X$: random variable $X$ with distribution $X \sim F(X)$ and density $f(x)$.

## B    ACQUISITION FUNCTIONS CONT.

**Random.** As a baseline we look at random acquisition. Random acquisition at cycle $k$ can be seen as uniformly sampling data from $\mathcal{D}_{\text{avail}}^k$:

$$\alpha_{\text{Random}}(\hat{g}^{k-1}(t), \mathcal{D}_{\text{avail}}^k) = \{t_1, \ldots t_b\} \sim \left\{ t_i; \frac{1}{n_{\text{avail}}} \right\}_{i=1}^{n_{\text{avail}}}. \tag{1}$$

Here, the acquisition function samples $b$ elements without replacement from the set of $n_{\text{avail}}$ elements. The set element $(t_i)$ is on the left of the semicolon, and the probability of the element being acquired $(\frac{1}{n_{\text{avail}}})$ is on the right of the semicolon. This convention will be used again below.

**BADGE.** BADGE looks to maximize the diversity of acquired samples, but, in contrast to Coreset, it additionally takes the uncertainty of the prediction into account (Ash et al., 2019). If the true label $y$ were observed, BADGE would proceed by maximizing the diversity of samples based on the gradient of the loss function $l$ with respect to the weights of the final layer of the most recently trained model $\tilde{w}^{k-1}$: $\frac{\partial}{\partial \tilde{\omega}^{k-1}} l(y, \hat{g}(t; \omega))$. Intuitively, it asks how much would our parameters change if we observed the labeled outcome for this example? However, the true label $y$ is not yet observed. Ash et al. (2019) explore BADGE in the classification setting. For a two class problem, where $f(y \mid t, \omega) = \text{Bernoulli}(y \mid \hat{g}(t; \omega))$, they propose using the class with the highest predicted probability, $\hat{y} = \text{argmax}_{y \in 0,1} f(y \mid t, \omega)$, to approximate the gradient as $\frac{\partial}{\partial \tilde{\omega}^{k-1}} l(\hat{y}, \hat{g}(t; \omega))$. This does not directly translate to the regression setting, as under our modelling assumptions the $y$ with the highest predicted likelihood corresponds exactly to $\hat{g}(t; \omega)$, which would lead to a loss of zero, and gradients of zero. As a starting point, we instead take $\hat{y}$ as a random sample from $f(y \mid t, \omega) = \mathcal{N}(y \mid \hat{g}(t; \omega), \sigma^2)$. We then use the same $k$-means++ algorithm as Ash et al. (2019) to approximate:

$$\alpha_{\text{BADGE}}(\hat{g}^{k-1}(t), \mathcal{D}_{\text{avail}}^k)$$

$$= \underset{\{t_1, \ldots t_b\} \in \mathcal{D}_{\text{avail}}^k}{\text{argmin}} \ \underset{t_i \in \mathcal{D}_{\text{avail}}^k}{\text{argmax}} \ \underset{t_j \in \mathcal{D}_{\text{avail}}^k \cup \mathcal{D}_{\text{cum}}^{k-1}}{\text{argmin}} \ \Delta \left( \frac{\partial l(\hat{y}, \hat{g}(t_i; \omega^{k-1}))}{\partial \tilde{\omega}^{k-1}}, \frac{\partial l(\hat{y}, \hat{g}(t_j; \omega^{k-1}))}{\partial \tilde{\omega}^{k-1}} \right) \tag{2}$$

where $\Delta$ is again the Euclidean distance.

**Bayesian Active Learning by Disagreement (BALD).** Given an uncertainty aware model, such as a BNN or Random Forest we can now take an information theoretic approach to selecting interventions from the pool data. Houlsby et al. (2011) frame active learning as looking to maximize the

information gain about the model parameters if we observe the outcome $Y = y$ given model inputs. Formally, the information gain is given by the mutual information between the random variables $Y$ and $\Omega$ given the intervention $t$ and acquired training data $\mathcal{D}_{\text{cum}}^{k-1}$ up until acquisition step $k$:

$$
\begin{aligned}
\mathcal{I}(Y; \Omega \mid t, \mathcal{D}_{\text{cum}}^{k-1}) &= H(Y \mid t, \mathcal{D}_{\text{cum}}^{k-1}) - H(Y \mid \Omega, t, \mathcal{D}_{\text{cum}}^{k-1}) \\
&= H(Y \mid t, \mathcal{D}_{\text{cum}}^{k-1}) - \mathbb{E}_{f(\omega \mid \mathcal{D}_{\text{cum}}^{k-1})} H(Y \mid \omega, t).
\end{aligned}
\tag{3}
$$

Under the assumed model we have

$$
\mathcal{I}(Y; \Omega \mid t, \mathcal{D}_{\text{cum}}^{k-1}) = \frac{1}{2} \log \left( \frac{\sigma^2 + \mathbb{E}_{f(\omega \mid \mathcal{D}_{\text{cum}}^{k-1})} \left[ \widehat{g}(t; \omega)^2 \right] - \mathbb{E}_{f(\omega \mid \mathcal{D}_{\text{cum}}^{k-1})} \left[ \widehat{g}(t; \omega) \right]^2}{\sigma^2} \right),
\tag{4}
$$

which leads to the following estimator setting $\sigma^2 = 1$

$$
\widehat{\mathcal{I}}(Y; \Omega \mid t, \mathcal{D}_{\text{cum}}^{k-1}) = \frac{1}{2} \log \left( 1 + \frac{1}{m} \sum_{j=1}^{m} \left( \widehat{g}(t; \omega_j^{k-1}) - \frac{1}{m} \sum_{j=1}^{m} \widehat{g}(t; \omega_j^{k-1}) \right)^2 \right).
\tag{5}
$$

We look at two acquisition functions for BALD. First, we consider the naive batch acquisition $\alpha_{\text{BALD}}$ proposed by Gal et al. (2017) which acquires the the top $b$ examples from $\mathcal{D}_{\text{avail}}^k$:

$$
\alpha_{\text{BALD}}(\widehat{g}^{k-1}(t), \mathcal{D}_{\text{avail}}^k) = \operatorname*{argmax}_{\{t_1, \dots t_b\} \in \mathcal{D}_{\text{avail}}^k} \sum_{i=1}^{b} \widehat{\mathcal{I}}(Y; \Omega \mid t_i, \mathcal{D}_{\text{cum}}^{k-1}).
\tag{6}
$$

This method will be referred to as `topuncertain` in the plots later. And second, we consider $\alpha_{\text{SoftBALD}}$ which randomly samples $b$ interventions from $\mathcal{D}_{\text{avail}}^k$ weighted by a tempered softmax function (Kirsch et al., 2021):

$$
\alpha_{\text{SoftBALD}}(\widehat{g}^{k-1}(t), \mathcal{D}_{\text{avail}}^k) = \{t_1, \dots t_b\} \sim \left\{ t_i; \frac{\exp \left( \frac{1}{\text{Temp}} \widehat{\mathcal{I}}(Y; \Omega \mid t_i, \mathcal{D}_{\text{cum}}^{k-1}) \right)}{\sum_{l=1}^{n_{\text{avail}}} \exp \left( \frac{1}{\text{Temp}} \widehat{\mathcal{I}}(Y; \Omega \mid t_l, \mathcal{D}_{\text{cum}}^{k-1}) \right)} \right\}_{i=1}^{n_{\text{avail}}},
\tag{7}
$$

where Temp $> 0$ is a user defined constant. This method will be referred to as `softuncertain` in the plots. As Temp $\to \infty$, $\alpha_{\text{SoftBALD}}$ will behave more like $\alpha_{\text{Random}}$. And as Temp $\to 0$, $\alpha_{\text{SoftBALD}}$ will behave more like $\alpha_{\text{BALD}}$.

**Coreset.** Coreset acquisition looks to maximize the diversity of acquired samples. This is done by finding the data points in $\mathcal{D}_{\text{avail}}^k$ that are furthest from the labelled data points in $\mathcal{D}_{\text{cum}}^{k-1}$. The robust K-centers algorithm of Sener & Savarese (2017) approximates a solution to:

$$
\alpha_{\text{CORESET}}(\widehat{g}^{k-1}(t), \mathcal{D}_{\text{avail}}^k) = \operatorname*{argmin}_{\{t_1, \dots t_b\} \in \mathcal{D}_{\text{avail}}^k} \operatorname*{argmax}_{t_i \in \mathcal{D}_{\text{avail}}^k} \operatorname*{argmin}_{t_j \in \mathcal{D}_{\text{avail}}^k \cup \mathcal{D}_{\text{cum}}^{k-1}} \Delta(t_i, t_j).
\tag{8}
$$

Euclidean distances, $\Delta(t_i, t_j)$, are calculated between the output of the penultimate layer of $\widehat{g}(t; \omega)$.

**Margin Sample.** Margin sampling is designed for classifiers where selection is based on the distance of a sample from the classifiers decision boundary (Roth & Small, 2006). As a proxy, the difference between the predicted probability of the most and second most probable classes is used. The distance between the most probable and the second most probable classes for a multi-class classification problem can be seen as how confident the model is about the label of that class. However, The concept of a decision boundary is ill-defined for regression tasks. One option to approximate margin sampling could be to model the aleatoric uncertainty of the model by predicting the conditional variance of the outcome $\sigma^2(t; \omega)$ and select data based on the magnitude of this value. Here, we instead look at the difference in the maximum and minimum values of the predicted outcome as a measure of the model's confidence and select data based on the magnitude of this value. Formally, we have

$$
\widehat{\mathcal{M}}(Y; \Omega \mid t_i, \mathcal{D}_{\text{cum}}^{k-1}) = \max_{j \in \{1, \dots m\}} (\widehat{g}(t; \omega_j^{k-1})) - \min_{j \in \{1, \dots m\}} (\widehat{g}(t; \omega_j^{k-1})),
\tag{9}
$$

and the acquisition function:

$$
\alpha_{\text{Margin}}(\widehat{g}^{k-1}(t), \mathcal{D}_{\text{avail}}^k) = \operatorname*{argmax}_{\{t_1, \dots t_b\} \in \mathcal{D}_{\text{avail}}^k} \sum_{i=1}^{b} \widehat{\mathcal{M}}(Y; \Omega \mid t_i, \mathcal{D}_{\text{cum}}^{k-1}).
\tag{10}
$$

Note that this approximation is similar to BALD under the assumption of a uniformly distributed outcome: $f(y \mid t, \omega) = \mathcal{U}(y \mid \widehat{g}(t; \omega))$.

**Adversarial Basic Iterative Method (AdvBIM).** Some of the adversarial algorithms can act as active learning acquisition functions by nominating the adversarial samples. Here, we extended the famous Adversarial BIM method for our regression task as an example. BIM was introduced by (Kurakin et al., 2016) to iteratively perturb adversarial samples to maximize the cost function $J$ subject to an $l_p$ norm constraint as

$$\hat{t}^{(0)} = t, \hat{t}^{(i)} = \text{clip}_{t,e}(\hat{t}^{(i-1)} + \text{sign}(\nabla_{\hat{t}^{(i-1)}} J(\theta, \hat{t}^{(i-1)}, y))) \tag{11}$$

(intermediate results are clipped to stay in $e$-neighbourhood of the primary data point $t$). This technique bypasses the intractable problem of finding the distance from the decision boundary by iteratively perturbing the features until crossing the boundary (Tramèr et al., 2017). In our regression task, we perturb the features in the gradients' direction to increase the conditional variance of the outcome, i.e.,

$$\hat{t}^{(0)} = t, \hat{t}^{(i)} = \text{clip}_{t,e}(\hat{t}^{(i-1)} + \text{sign}(\nabla_{\hat{t}^{(i-1)}} \text{Var}_\omega(\hat{g}(t,\omega)))) \text{ for } i = \{1, \ldots, m\}, \tag{12}$$

where $||\hat{t}_i - t||_2 < \gamma * ||t||_2$ with the hyperparameter $\gamma$. After creating adversarial samples for each data point in $D_{\text{avail}}^k$, $\alpha_{\text{AdversarialBIM}}$ acquires the samples by

$$\alpha_{\text{AdversarialBIM}}(\hat{g}^{k-1}(t), \mathcal{D}_{\text{avail}}^k) = \bigcup_{t_i \in \mathcal{D}_{\text{avail}}^k} \underset{t_j \in \mathcal{D}_{\text{avail}}^k}{\text{argmin}} \Delta(\hat{t}_i^{(m)}, t_j), \tag{13}$$

where $\Delta$ is the euclidean distance.

$k$**-means Sampling.** This method nominates samples by returning the closest sample to each center of the unlabeled data clusters. In order to do so, one may run Kmeans++ clustering algorithm with the number of clusters equal to $b$ over either the unlabeled data points $D_{\text{avail}}^k$ or the output of the penultimate layer of $\hat{g}(t; \omega)$. We refer to the former as `kmeansdata` and to the latter as `kmeansembed` in the experiments. Assuming $\{\mu_1, \ldots, \mu_b\}$ are the centers of the clustering, we have

$$\alpha_{\text{Kmeans}}(\hat{g}^{k-1}(t), \mathcal{D}_{\text{avail}}^k) = \bigcup_{i=1}^{b} \underset{t_j \in \mathcal{D}_{\text{avail}}^k}{\text{argmin}} \Delta(\mu_i, t_j), \tag{14}$$

where $\Delta$ is euclidean distance over the data points or the penultimate layer of $\hat{g}(t; \omega)$.

## C DETAILED EXPERIMENTAL RESULTS

### C.1 BAYESIAN NEURAL NETWORK (BNN) MODEL

We provide here detailed experimental results across all hyperparameter settings. The result of fig. 2 that was presented for 3 batch sizes are provided for 6 batch sizes in fig. 4. Similarly, the results of fig. 3 are provided for additional batch sizes in fig. 7. In addition, both fig. 2 and fig. 3 report the results for the STRING treatment descriptors. All experiments are repeated for two other sets of input treatment descriptors (Achilles and CCLE) whose results are provided in figs. 5, 6, 8 and 9.

### C.2 RANDOM FOREST MODEL

In addition to the BNN model, we carried out thorough analyses for a different model class. The experiments are repeated for the random forest as an uncertainty aware ensemble model. The uncertainty in random forests, similar to other ensemble methods, is originated from the prediction made by each model instance in the ensemble. We use the random forest implementation in the Scikit-learn package (Pedregosa et al., 2011) with 100 trees and set the option max_depth=None so that the depth of the trees are determined automatically. The performance of the model trained over the active learning cycles can be seen in fig. 10 for different acquisition functions, different batch sizes, different target datasets, and the STRING treatment descriptors. Similarly, the hit ratio of the interesting genes for a random forest model is reported in fig. 12. The same experiment was repeated for CCLE treatment descriptors whose results are provided in fig. 11 and fig. 13. Notice that random forest experiments are done with a reduced set of acquisition functions that could be adjusted to the random forest model.

### C.3 IN-DEPTH DESCRIPTION OF THE *Hit Ratio* EXPERIMENT

Here we elaborate more on the purpose and the message of the hit ratio experiment whose results are reported in figs. 7 to 9, 12 and 13 for various settings. The purpose of these experiments is to compare the performance of different acquisition functions in different settings of batch sizes and input/output datasets to hit the gene targets that are known to be interesting by genomics experts. To choose the set of interesting genes, we sort them based on their absolute target values. Then we choose the top $5\%$ of this list that corresponds to both extremes of positive and negative values (both extremes are considered to be good targets by experts.) The experiments are repeated for 5 different random seeds to obtain the error bars.

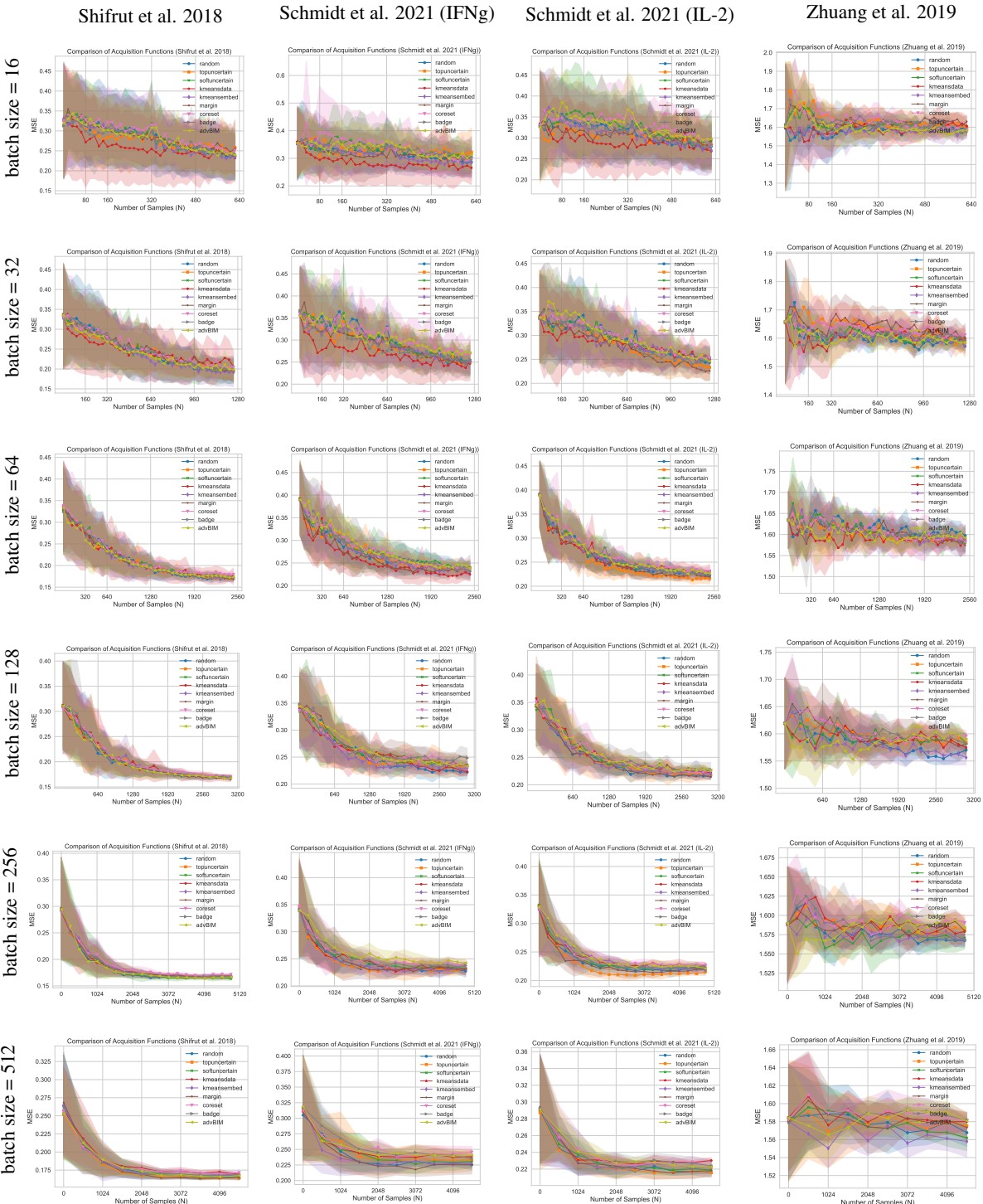

Figure 4: The evaluation of the model trained with STRING treatment descriptors at each active learning cycle for 4 datasets and 6 acquisition batch sizes. In each plot, the x-axis is the active learning cycles multiplied by the acquisition bath size that gives the total number of data points collected so far. The y-axis is the test MSE error evaluated on the test data.

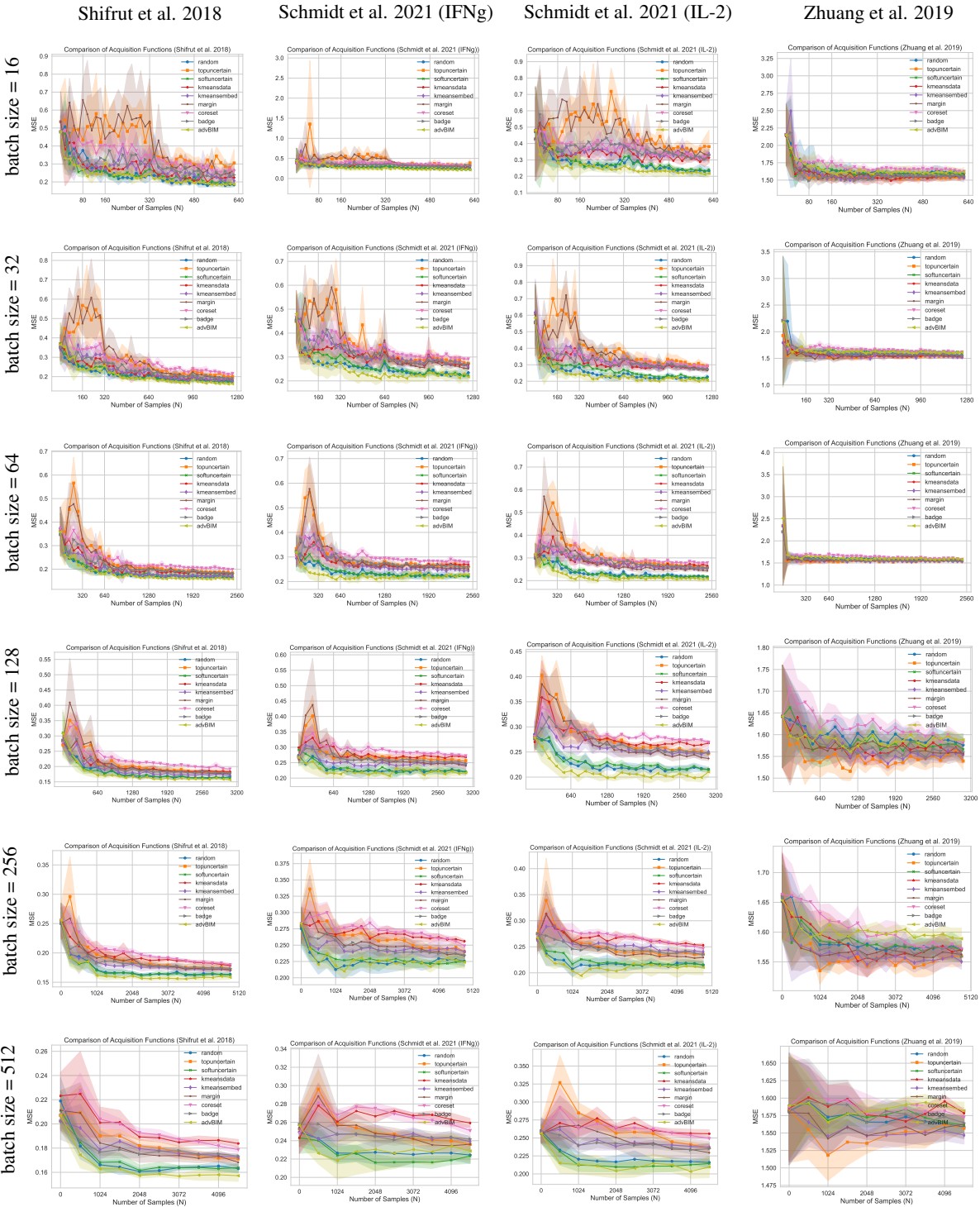

Figure 5: The evaluation of the model trained with Achilles treatment descriptors at each active learning cycle for 4 datasets and 6 acquisition batch sizes. In each plot, the x-axis is the active learning cycles multiplied by the acquisition bath size that gives the total number of data points collected so far. The y-axis is the test MSE error evaluated on the test data.

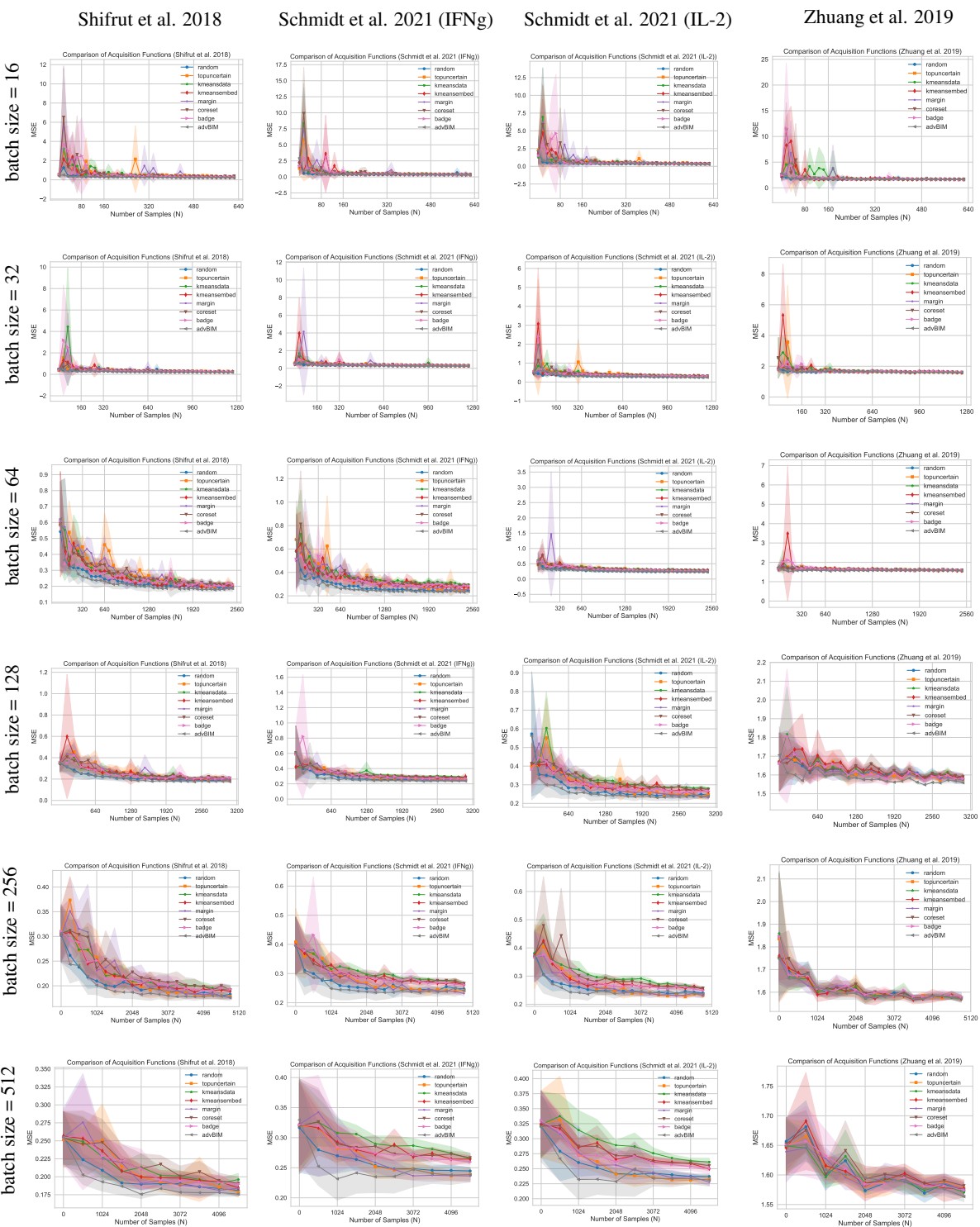

Figure 6: The evaluation of the model trained with CCLE treatment descriptors at each active learning cycle for 4 datasets and 6 acquisition batch sizes. In each plot, the x-axis is the active learning cycles multiplied by the acquisition bath size that gives the total number of data points collected so far. The y-axis is the test MSE error evaluated on the test data.

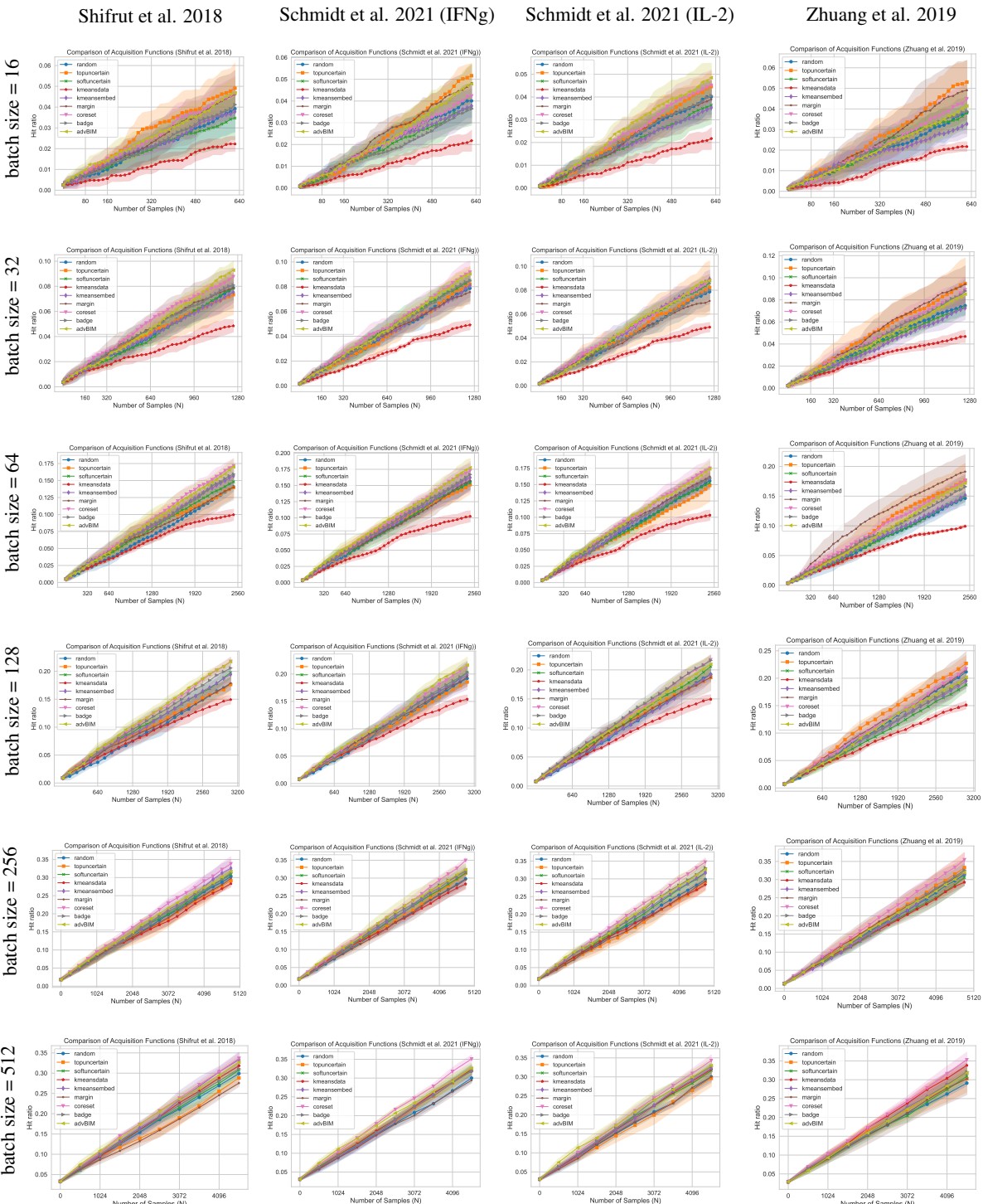

Figure 7: The hit ratio of different acquisition for BNN model, different target datasets, and different acquisition batch sizes. We use STRING treatment descriptors here. The x-axis shows the number of data points collected so far during the active learning cycles. The y-axis shows the ratio of the set of interesting genes that have been found by the acquisition function up until each cycle.

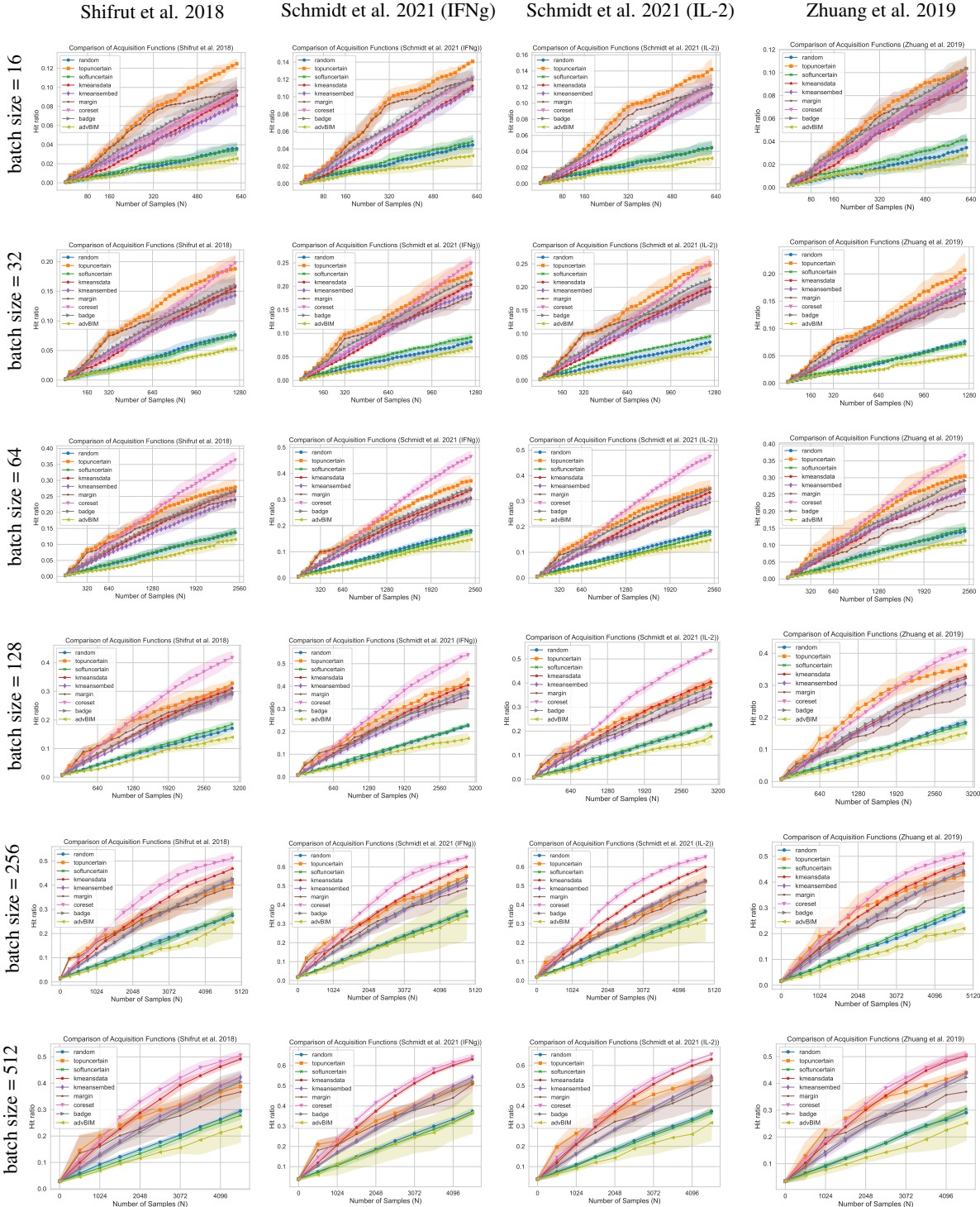

Figure 8: The hit ratio of different acquisition for BNN model, different target datasets, and different acquisition batch sizes. We use Achilles treatment descriptors here. The x-axis shows the number of data points collected so far during the active learning cycles. The y-axis shows the ratio of the set of interesting genes that have been found by the acquisition function up until each cycle.

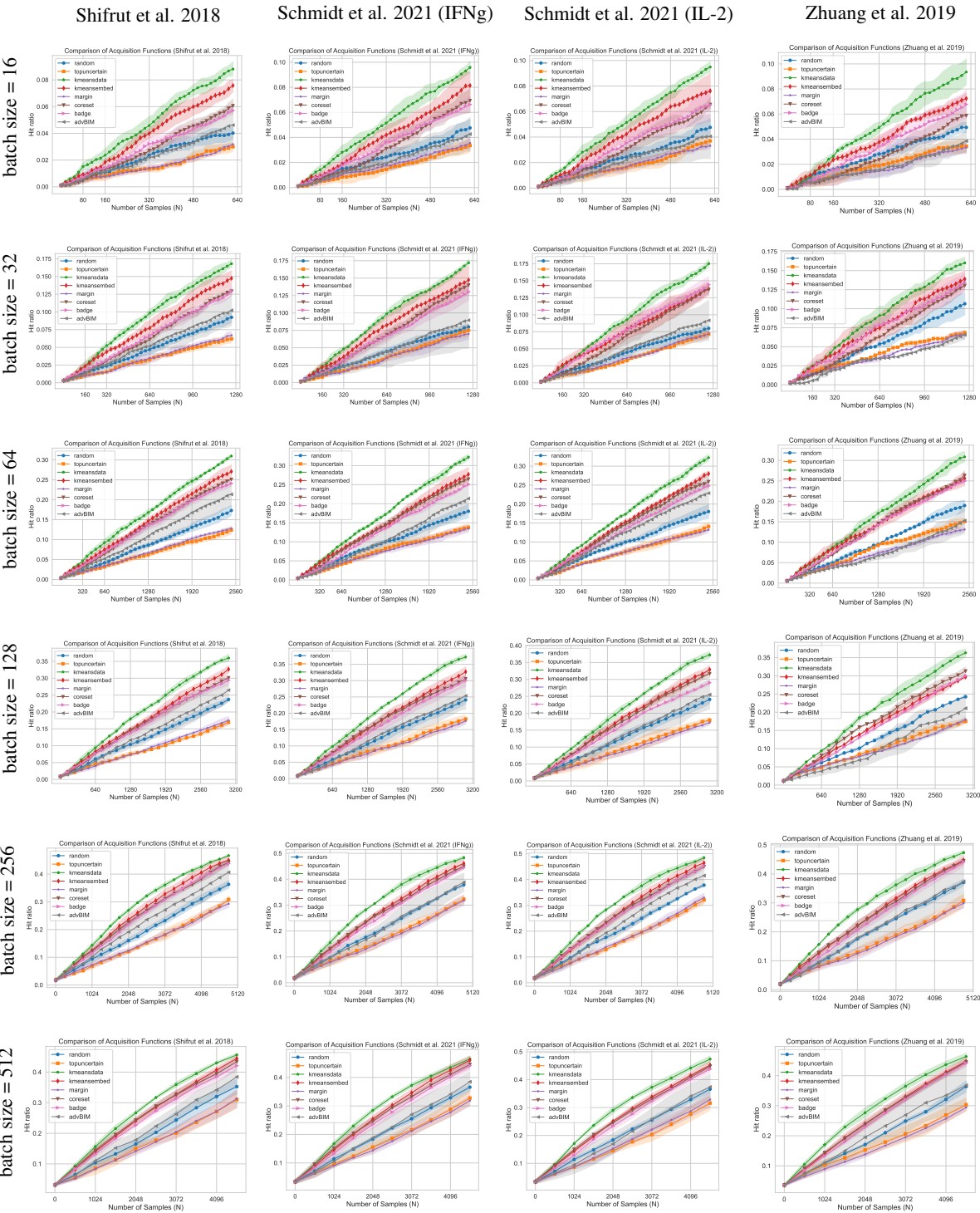

Figure 9: The hit ratio of different acquisition for BNN model, different target datasets, and different acquisition batch sizes. We use CCLE treatment descriptors here. The x-axis shows the number of data points collected so far during the active learning cycles. The y-axis shows the ratio of the set of interesting genes that have been found by the acquisition function up until each cycle.

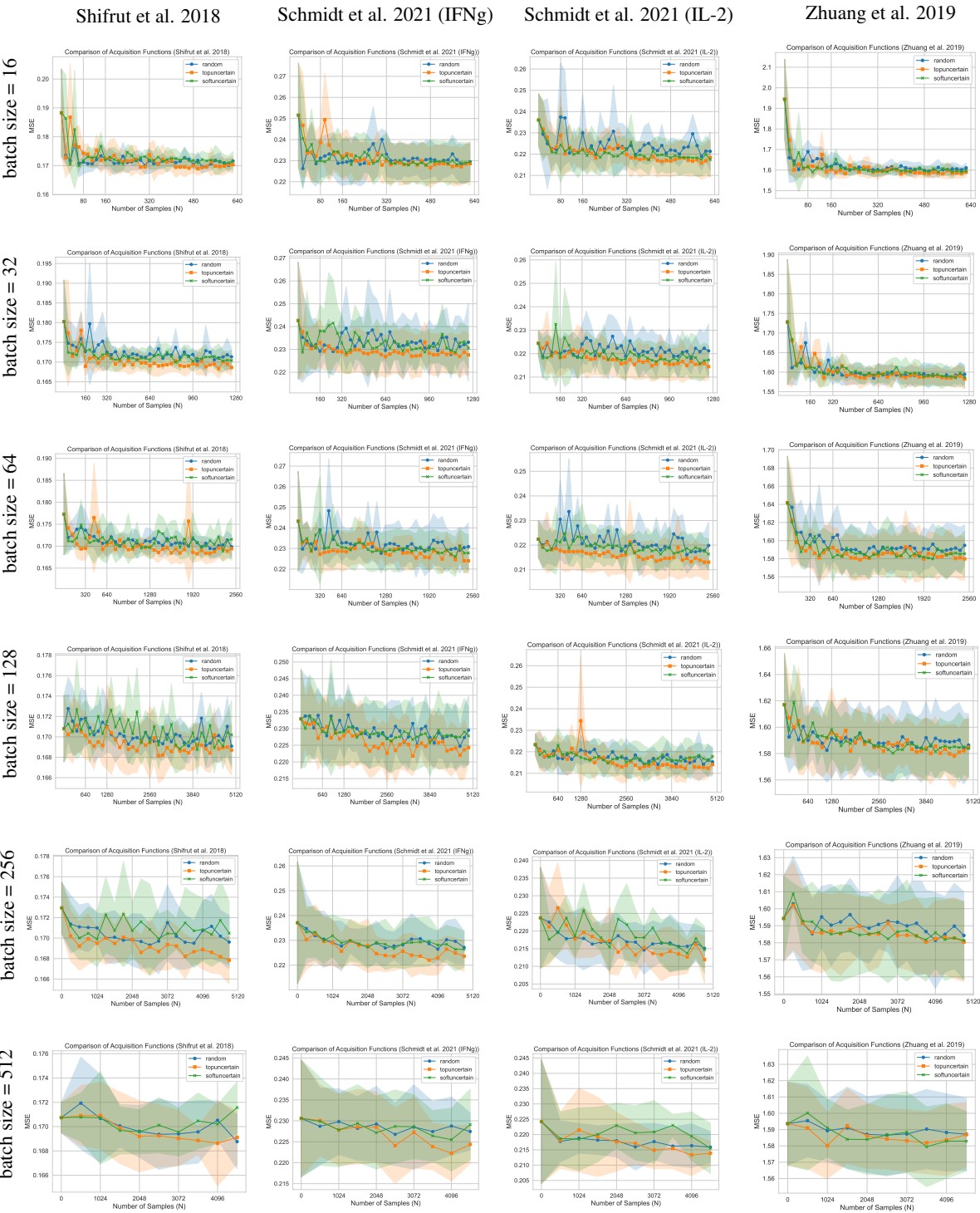

Figure 10: The evaluation of the random forest model trained with STRING treatment descriptors at each active learning cycle for 4 datasets and 6 acquisition batch sizes. In each plot, the x-axis is the active learning cycles multiplied by the acquisition bath size that gives the total number of data points collected so far. The y-axis is the test MSE error evaluated on the test data.

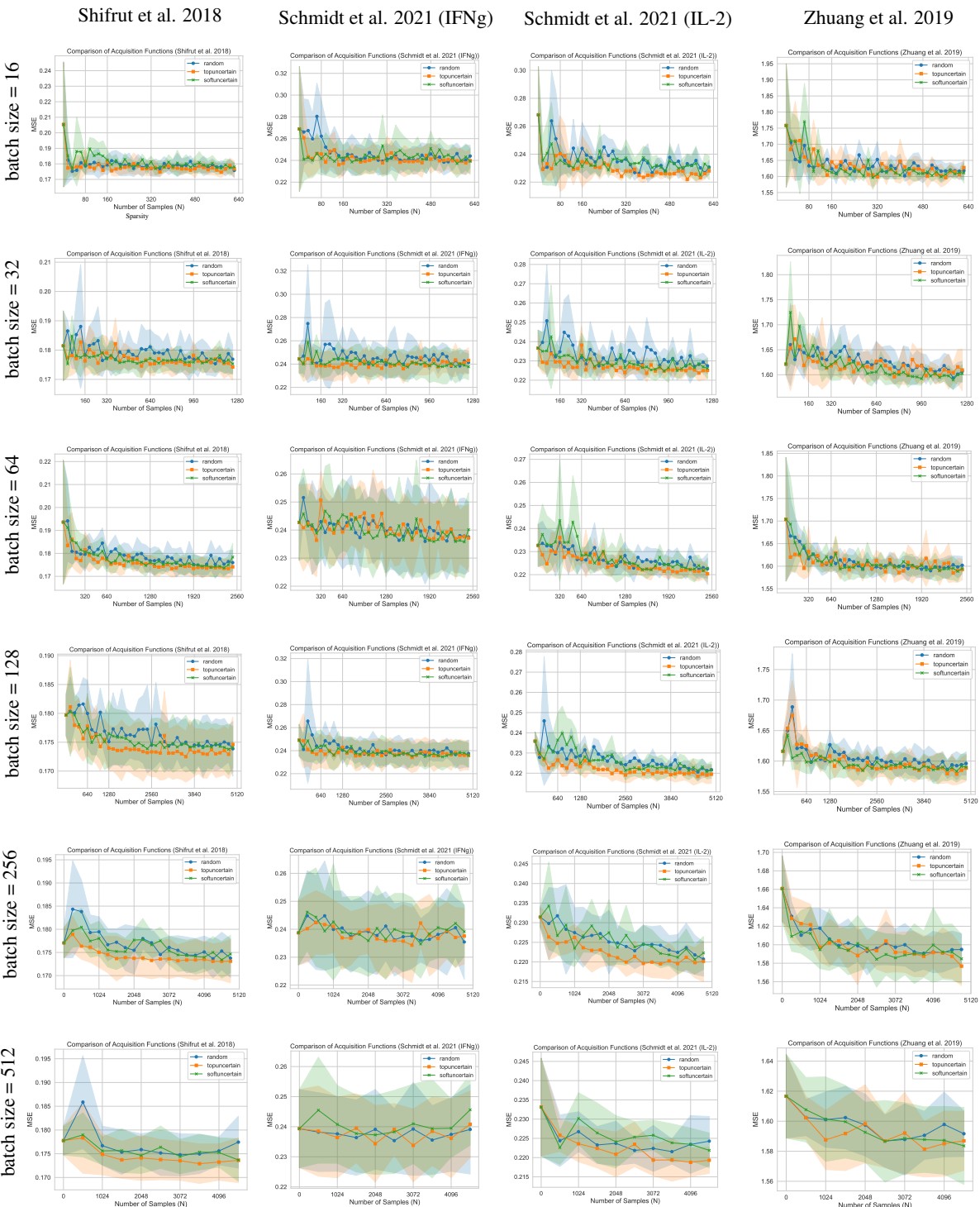

Figure 11: The evaluation of the random forest model trained with CCLE treatment descriptors at each active learning cycle for 4 datasets and 6 acquisition batch sizes. In each plot, the x-axis is the active learning cycles multiplied by the acquisition bath size that gives the total number of data points collected so far. The y-axis is the test MSE error evaluated on the test data.

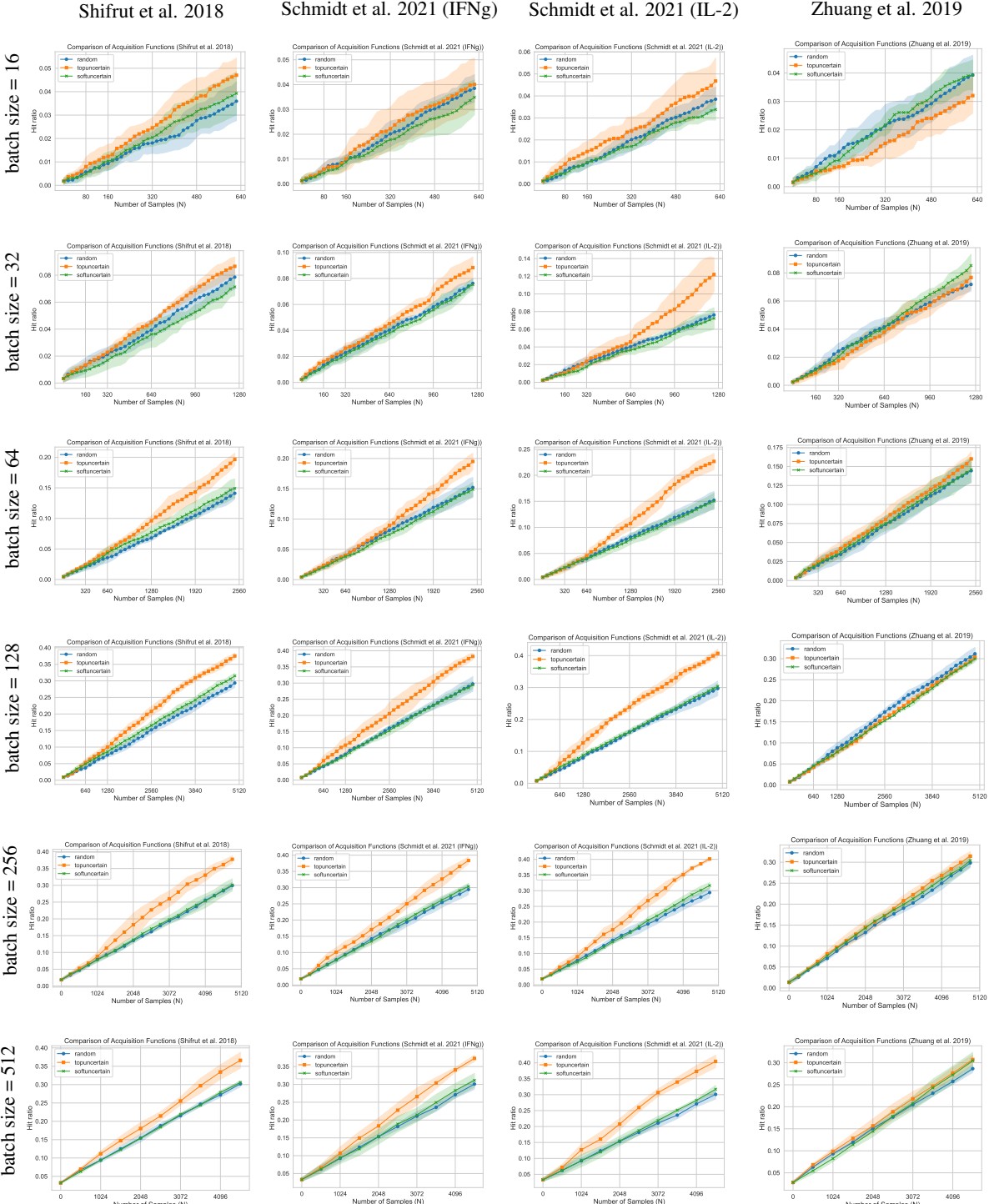

Figure 12: The hit ratio of different acquisition for random forest model, different target datasets, and different acquisition batch sizes. We use STRING treatment descriptors here. The x-axis shows the number of data points collected so far during the active learning cycles. The y-axis shows the ratio of the set of interesting genes that have been found by the acquisition function up until each cycle.

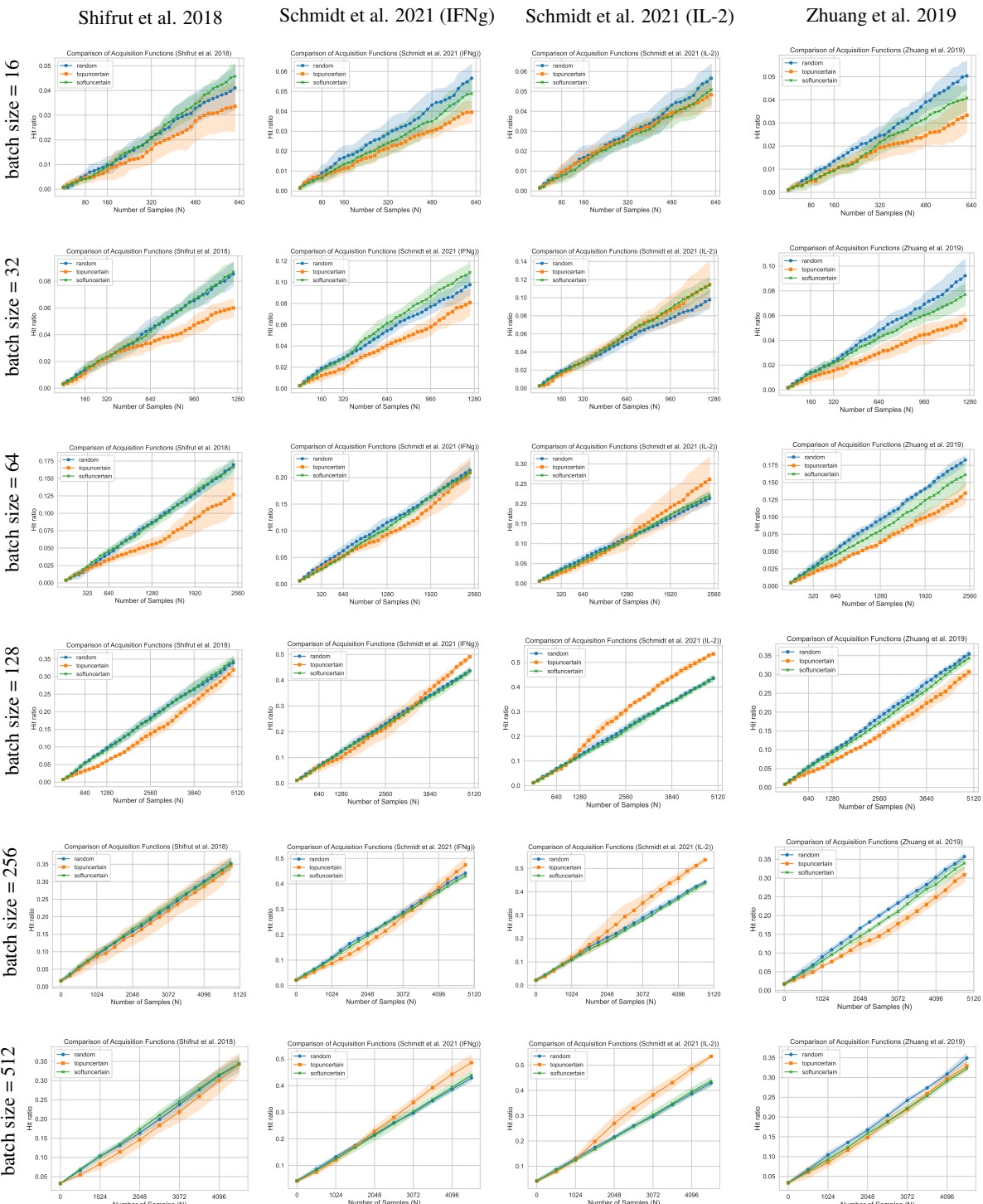

Figure 13: The hit ratio of different acquisition for random forest model, different target datasets, and different acquisition batch sizes. We use CCLE treatment descriptors here. The x-axis shows the number of data points collected so far during the active learning cycles. The y-axis shows the ratio of the set of interesting genes that have been found by the acquisition function up until each cycle.

# D  ADDITIONAL DATASETS

We tested the GeneDisco pipeline on two additional datasets which are included in this appendix due to space constraints. The relative performance of the random forest model with the acquisition functions that are compatible wit this model class are shown in Figure 14 for the predictive accuracy task and Figure 15 for the hit ratio task.

## D.1  MODULATION OF TAU PROTEINS IN NEURONS

**Experimental setting.**  This assay is based on Sanchez et al. (2021) in which authors have conducted genome-wide CRISPR screens in two SH-SY5Y neuroblastoma cell lines to identify genes that, when knocked out, either increased or decreased expression of endogenous tau proteins.

**Measurement.**  After editing (for 21 or 30 days), cells are FACS sorted based on low tau expression (low quartile 25%) and high tau expression (high quartile 25%). Statistical significance of gRNA enrichment is determined via a redundant siRNA activity (RSA; log p-value) analysis. RSA up scores were used to find genes enriched in the 25% of cells with highest tau protein, while RSA down scores were used to find genes enriched in the 25% of cells with the lowest tau protein.

**Importance.**  While the exact mechanism leading to the buildup of Tau proteins is unknown, their accumulation is correlated with several neurodegenerative pathologies (eg., Alzheimer's disease, progressive supranuclear palsy, and frontotemporal dementia). The genes identified in this screen may therefore help gaining a better understanding of the underlying disease pathways as well as leading to potential treatments.

## D.2  REGULATION OF ENDOSOMAL ENTRY IN CELLS FOR SARS-COV-2

**Experimental setting.**  This dataset is based on the genome-wide CRISPR screen described in Zhu et al. (2021). The assay was designed to identify endosomal entry-specific regulators of SARS-CoV-2 virions in A549-ACE2 cells.

**Measurement.**  The top candidates from the CRISPR screen were determined according to their MAGeCK score (- log10).

**Importance.**  The endosomal pathway is one of the two pathways (along with fusion at the plasma membrane) used by SARS-CoV-2 to infect cells. A better understanding of mechanisms underpinning this pathway may help identify targets for the development of new SARS-CoV-2 antiviral therapeutics.

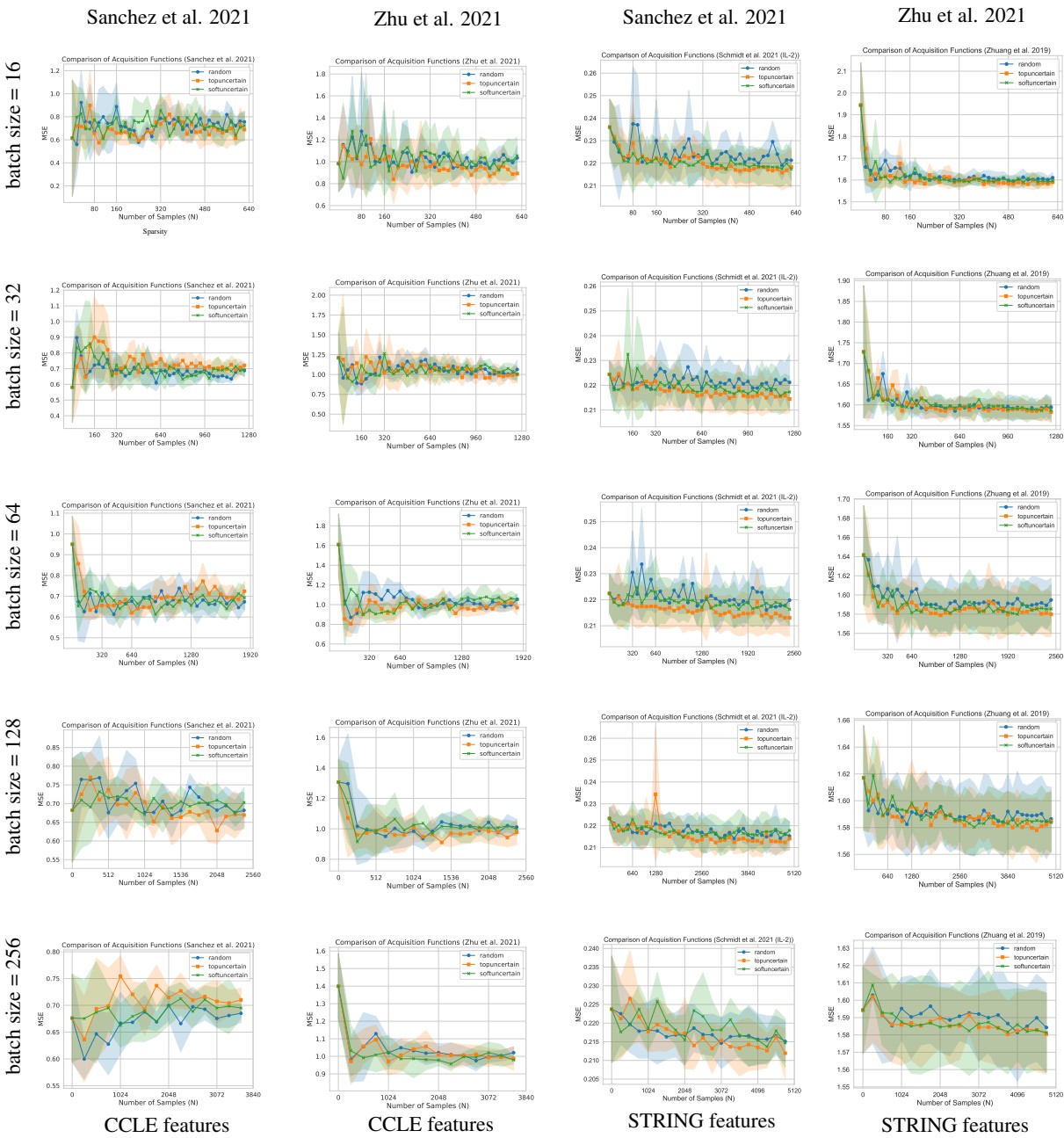

Figure 14: The evaluation of the random forest model trained with CCLE and STRING treatment descriptors at each active learning cycle for the datasets (Sanchez et al., 2021) and (Zhu et al., 2021) and also for 5 acquisition batch sizes. In each plot, the x-axis is the active learning cycles multiplied by the acquisition bath size that gives the total number of data points collected so far. The y-axis is the test MSE error evaluated on the test data.

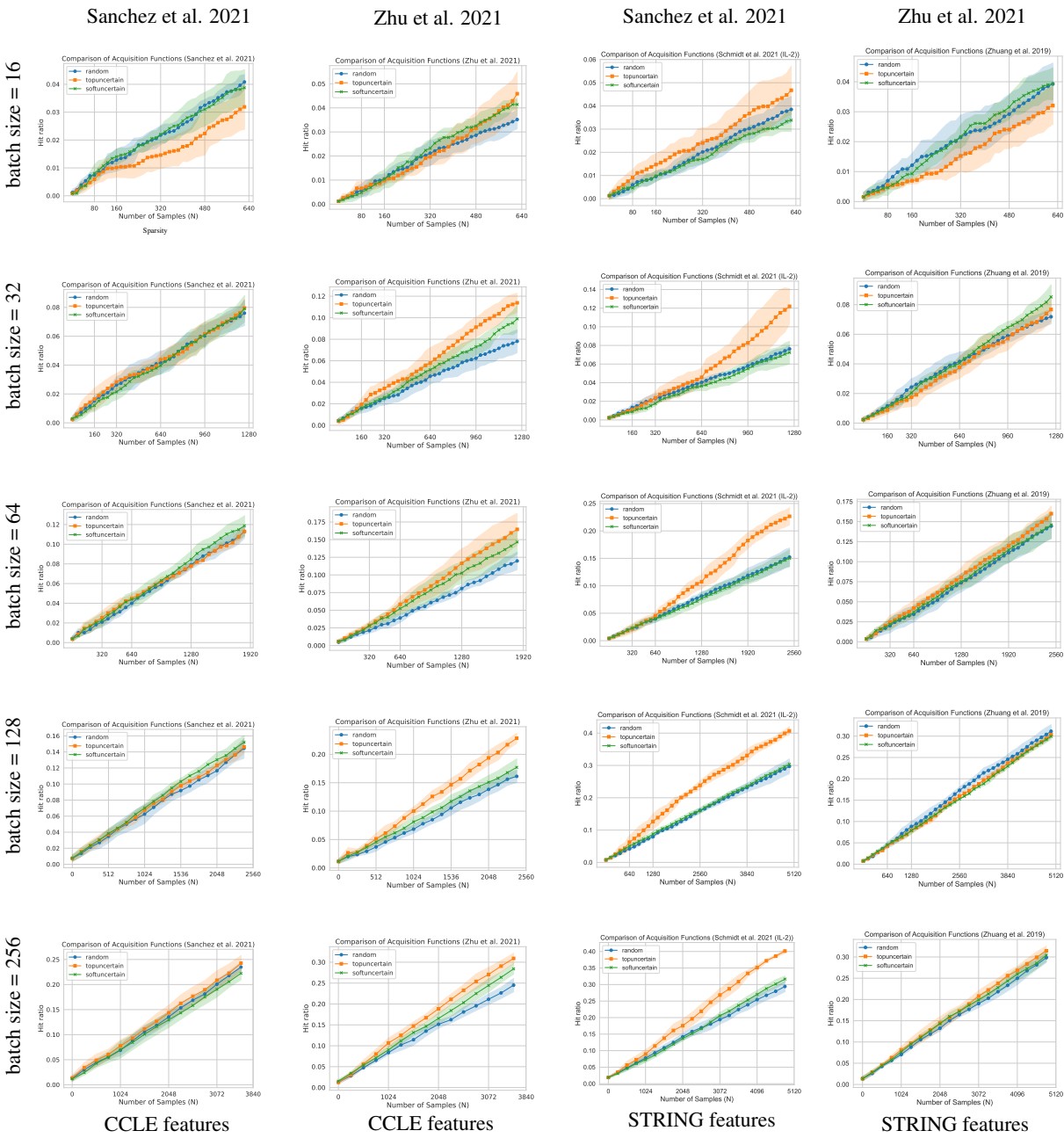

Figure 15: The hit ratio of different acquisition for random forest model, different target datasets, and different acquisition batch sizes. We use STRING and CCLE treatment descriptors here. The x-axis shows the number of data points collected so far during the active learning cycles. The y-axis shows the ratio of the set of interesting genes that have been found by the acquisition function up until each cycle.

