# OpenReview forum: "GeneDisco: A Benchmark for Experimental Design in Drug Discovery"
_ICLR.cc/2022/Conference — ICLR 2022 Poster_

### Official Review · Reviewer_6EvA · 2021-10-26

**Correctness:** 3
**Technical Novelty And Significance:** 1
**Empirical Novelty And Significance:** 3
**Recommendation:** 6
**Confidence:** 4

**Main Review:**

Pros:
- I feel this kind of benchmark would be useful. Although the  data sets are already publicly available, it's nevertheless nice if someone else has done the required work and one can start modeling immediately.
- The presentation is mostly clear.

Cons/questions:
- The paper is straightforward, does not have any particular shortcomings, and is potentially useful to the community. However, I feel it lacks the kind of machine learning novelty and significance that would be expected from an ICLR publication, and hence some other forum could be more suitable (though I'm happy to discuss this with the other reviewers and the area chair).
- Figures 2 and 3 require a lot of zooming in before they are readable, and they are not at all legible in a printed-out version.
- I found the discussion on the comparison of acquisition methods a bit superficial.

**Summary Of The Paper:**

The paper presents a benchmark for experimental design in drug discovery based on previously public data sets. The benchmark includes three standardized feature sets describing possible interventions and four different genome-wide CRISPR experiment results (from which the counterfactual outcomes for an intervention on each gene are obtained). In addition, the paper uses the benchmark to compare different active learning strategies (nine acquisition functions in total), where the goal is to improve prediction accuracy and hit ratio of interesting genes among the interventions selected by a specific (batch) acquisition function.


**Summary Of The Review:**

A straightforward paper that presents a useful-looking benchmark dataset for drug discovery. Machine learning novelty and significance is limited.

POST-REBUTTAL UPDATE:
After clarifications, it seems apparent that benchmark papers like this could fall in the scope of ICLR. I still find the benchmark potentially useful and something I might use in my own research. I do not have any particular shortcomings in mind, but I think a stronger demonstration of the ability to generate new machine learning insights using the benchmark would have been useful. I will update my score to borderline positive (6) to reflect these thoughts.

---

> ### Author Response · Authors · 2021-11-21
> **Response to Reviewer 6EvA's concerns**
>
> Thank you very much for taking the time to review our work and recognize its merits. We agree that our work will facilitate the access of different research teams to various datasets in a clean and standardized format, and also believe that it will contribute to ensuring consistency in assessing results from various teams.
> (AR: Author Response)
>
> ---
>
> **The paper is straightforward, does not have any particular shortcomings, and is potentially useful to the community. However, I feel it lacks the kind of machine learning novelty and significance that would be expected from an ICLR publication, and hence some other forum could be more suitable (though I'm happy to discuss this with the other reviewers and the area chair).**
>
> AR: Please see our message to the AC on that point (“Guidance on how to review benchmark papers”): since this work is a benchmark paper, it has different objectives compared with other methodology-focused papers and is based on the ICLR reviewer guidelines (point 2.1), it should be assessed as such. In particular, prior benchmark papers accepted at ICLR in the past 3 years did not introduce new methodology in the areas they were focused on. We believe novelty for this type of work should be assessed with respect to the prior existence of other benchmarks in the area, and to our knowledge, there is no such benchmark, neither for biological experiment design nor more broadly for active learning in general. Consequently, we believe our work addresses an important gap that will be useful for several machine learning sub-communities.
>
> ---
>
> **Figures 2 and 3 require a lot of zooming in before they are readable, and they are not at all legible in a printed-out version.**
>
> AR: Thank you for the feedback -- we will increase the font size to facilitate the reading of printouts.
>
> ---
>
> **Found the discussion on the comparison of acquisition methods a bit superficial.**
>
> AR: There is a detailed discussion of different methods in Appendix B (we also added a general discussion on different method types in the main text) of the original manuscript (due to space constraints). Moreover, we added a more in-depth discussion about the interplay between the acquisition function and the predictive model in Section 6 (Discussion and Conclusion) and of the revised manuscript. Please let us know if there is anything else to clarify about different methods.
>
> ---

---

### Official Review · Reviewer_ozjG · 2021-11-02

**Correctness:** 3
**Technical Novelty And Significance:** 3
**Empirical Novelty And Significance:** 4
**Recommendation:** 6
**Confidence:** 4

**Main Review:**

The strengths of this paper include :

-  The paper focuses on an important aspect of experimental design for drug discovery, which has not previously been studied. This type of benchmarking is invaluable to make the best possible use of limited experimental resources.

- The datasets are carefully designed with input from researchers in genomics and CRISPR screens, minimizing potential sources of error (e.g., no duplication).

- It has the right level of detail; it makes specific recommendations that can be readily implemented by developers or researchers interested in working with similar data sets.

- Source code is made publicly available online.

The limitations include:

- The benchmarks contain relatively little information about how far apart genes are located (i.e., there is no distance metric). In practice, this information can be very useful for active learning algorithms.

- The authors study only two candidate algorithms so far; however it is still an important step forward because the novel dataset and methodology enable other groups to conduct extensive evaluations on their own machine learning algorithms.

- The problem formulation should be clarified and is difficult to understand as there are multiple different datasets presented.

- A thorough justification for the assumptions utilized in the parametric models; specifically the outcome Y, should be disclosed.

- How do the authors define the ground truth when evaluating the active learning strategies?

- How do the authors define hits for the different active learning strategies?

- The results show rather uniform performance regardless of the active learning strategy employed. Can the authors comment on the reasoning around this phenomenon and whether it is an artifact of the problem formulation or datasets employed?

- It is unclear if biological replicates are included in the experiments. A change in this fundamental would have a very substantial impact on experimental design benchmarks and biology interpretations. If replicates were not used, they should be added to the text.

**Summary Of The Paper:**

The authors propose a benchmarking suite for evaluating active learning strategies to experimental design in drug discovery. The authors focus on active learning for in vitro genetic experiments, where the goal is to identify which genes or proteins underlie a certain disease (e.g., by using CRISPR technologies). For this task, there are typically billions of potential hypotheses to test. The experimental design space for such in vitro cellular experiments is extremely vast; thus, even with help from modern computers and machine-learning algorithms (which can search through the entire space), it remains practically impossible to exhaustively test every possible hypothesis that lives in this biological hypothesis space.

The paper contributes a benchmarking suite that can be used to evaluate how well active learning strategies are able to learn useful experimental designs. The paper has the following contributions:

- A new experimental design benchmark suite of genetic experiments for CRISPR screens with unique challenges not found in other benchmarks. This includes demonstrating cost-efficiency by using only a small number of experiments relative to the total number of hypotheses tested.

- An evaluation methodology for experimental designs on GeneDisco data sets, including methods for evaluating cost efficiency and quality of information gain during active learning.

- Initial evaluations apply this approach to two candidate causal inference algorithms: one based on a Bayesian neural network and another based on a Random Forest. Further methodological development is required in order to explore more sophisticated algorithms for active learning.

- The paper is published with supplementary material, including the two experimental design benchmark datasets and all source code necessary to reproduce the results of this paper.

**Summary Of The Review:**

All in all, this paper proposes a benchmarking suite that can help developers of active learning algorithms improve their programs. While the paper only considers a limited set of models, active learning algorithms, and datasets, it is an important step towards establishing best practices for in vitro experimental design. With source code publicly available online, it provides necessary tools (and datasets) to advance the state of the art for experimental design in drug discovery studies that use CRISPR technologies.

---

> ### Author Response · Authors · 2021-11-19
> **Response to Reviewer ozjG's concerns (ctd)**
>
> **A thorough justification for the assumptions utilized in the parametric models; specifically the outcome Y, should be disclosed.**
>
> AR: The estimation of the map from gene intervention to phenotype screens is based on the assumption that the intervention is the cause of the relative change in the assay, hence is predictive of such change.
>
> Given that the interventional outcomes are continuous we believe that the assumption of Gaussian likelihood in the model output is standard and a good starting point and serves as a sensible baseline. We clarified this assumption in Section 4.3. and leave it to future work to explore heterogeneity in output noise and other likelihood functions.
>
> ---
>
> **How do the authors define the ground truth when evaluating the active learning strategies?**
>
> AR: We outline the definition of the ground truth interventional outcomes for each assay in section 4.2. Could you please let us know if there is any particular aspect of the ground truth definition that you felt was missing?
>
> ---
>
> **How do the authors define hits for the different active learning strategies?**
>
> AR: We define hits in section C3 of the appendix, we can move this to the main text if you think this would enhance the quality of the paper.
>
> ---
>
> **The results show rather uniform performance regardless of the active learning strategy employed. Can the authors comment on the reasoning around this phenomenon and whether it is an artifact of the problem formulation or datasets employed?**
>
> AR: While the uncertainty bounds for several acquisition functions are overlapping in terms of the Mean Squared Error (MSE) model performance metric (Fig. 2), we observed significant differences between acquisition functions when evaluated in terms of the ratio of hits discovered given the same number of cycles (Fig. 3). For example, in the Shifrut et al. prediction task at batch size 16 `kmeansdata` achieved a maximum of slightly more than 2% of interesting hits discovered, whereas `topuncertain`discovered almost 5% of the interesting genetic targets (Fig. 3) - corresponding to a more than two-fold increase in discovered targets using the better acquisition strategy. Similar differences can be seen for a range of the prediction tasks we evaluated and we therefore believe that our results do indicate that the search for better acquisition functions can yield considerable improvements in terms of overall discovery rates of interesting gene targets.
>
> ---
>
> **It is unclear if biological replicates are included in the experiments. A change in this fundamental would have a very substantial impact on experimental design benchmarks and biology interpretations. If replicates were not used, they should be added to the text.**
>
> AR: Thank you very much for your feedback on this point, and we will clarify that point in the revised version. Biological replicates are available for a subset of the assays in the benchmark: the two assays in Schmidt et al. are carried out on two independent donors, and we use the mean response across donors in experiments to train models on a more robust signal. With that said, there will be many practical situations where one has to rely on models trained on noisy data. While more reliable biological measurements are certainly helpful to train models that will drive more solid conclusions for a fixed experiment, we believe that having noisier assays in our benchmark (e.g., with no biological nor technical replicates) will also support the development of new algorithms that are more suited to noisy data regimes. Consequently, having a balance between noisy and less-noisy assays (with biological replicates) in our benchmark helps compare how different algorithms fare in the face of different noise regimes.

---

> ### Author Response · Authors · 2021-11-19
> **Response to Reviewer ozjG's concerns**
>
> Thank you for enumerating the strengths of our work and recognizing “its value in making the best use of limited experimental resources and presenting the right level of details”. We address each of your concerns in the following and look forward to further discussion if you have additional feedback.
> (AR: Author Response)
>
> ---
>
> **The benchmarks contain relatively little information about how far apart genes are located (i.e., there is no distance metric). In practice, this information can be very useful for active learning algorithms.**
>
> AR: The distance between genetic intervention descriptors, T, is determined by their functional similarity. In Section 4.1 we describe how this functional similarity is determined for each of the treatment descriptor sets we provide, achilles, STRING, and CCLE. The descriptors are dense continuous spaces with different dimensionalities (achilles: 808, STRING: 799, CCLE: 420). Suitable distance metrics in these spaces could be Mahalanobis distance or cosine similarity. We would be interested to know how you would recommend communicating the distance between genes? Are you interested in the average distance across all descriptors, or something that communicates pairwise distances?
>
> ---
>
> **The authors study only two candidate algorithms so far; however it is still an important step forward because the novel dataset and methodology enable other groups to conduct extensive evaluations on their own machine learning algorithms.**
>
> AR: We provide results for two separate models to show that the uniform performance for model error across the nine different active learning strategies that you have pointed out is not just an artifact of a given model. We agree that is still an important step forward and hope these initial results motivate others to use the benchmark and explore the use of different models for this problem.
>
> ---
>
> **The problem formulation should be clarified and is difficult to understand as there are multiple different datasets presented.**
>
> AR: We think your comment here points to an improvement we are happy to make. It seems that we can improve the connection between the problem setting in Section 3, where we define the mathematical definitions of the intervention descriptors T and intervention outcome Y, with the dataset definitions in section 4, where we describe the different intervention descriptors (4.1) and intervention outcomes (4.2)  provided in the proposed GeneDisco benchmark. Do you have concrete suggestions on how we could best communicate this?
>
> ---

---

### Official Review · Reviewer_NhJz · 2021-11-03

**Correctness:** 3
**Technical Novelty And Significance:** 2
**Empirical Novelty And Significance:** 2
**Recommendation:** 6
**Confidence:** 3

**Main Review:**

Strengths
1) The authors are commended for addressing the hard problem of causal inference in biological systems.
2) The authors give a good explanation of the methodology, datasets, and metrics.
3) A good explanation of the genetic assays are provided

Weakness:
1) The title is misleading. The paper is much more about gene knockdown than drug discovery. As presented, causal connections between specific loss-of-function mutations and immunologic phenotypes may be drawn, but drug discovery is not directly related.
2) The authors do not provide any novel methodologies within the field of active learning or drug development.
3) The assays are limited to a small subset of immunologic phenotypes.


**Summary Of The Paper:**

The authors introduce a benchmark suite for evaluating active learning algorithms for experimental design in drug discovery. Specifically, they introduce various genome-wide CRISPR screens within immunology that evaluate the causal effect of intervening on a large number of genes in model systems in order to identify the genetic perturbations that induce a desired phenotype. The work focuses on using counterfactual estimators of experimental outcomes to propose experimental hypotheses for validation in in vitro experiments with genetic interventions (CRISPR) in order to discover potential causal associations between biological entities that could be relevant for the development of novel therapeutics.

The authors standardize two types of datasets: three standardized feature sets describing interventions and four difference in vitro genome-wide CRISPR experimental assays. Two model types are used (BNN and Random Forest Regression) and nine different acquisition functions.

Results are presented in Figures 2 and 3.


**Summary Of The Review:**

Overall, the authors provide a decent dataset and benchmark for genetic knockdown experimentation in the setting of active learning. However, I believe the paper would be better suited for datasets/benchmarks venue.

---

> ### Author Response · Authors · 2021-11-19
> **Response to Reviewer NhJz's concerns**
>
> Thank you for the constructive and helpful comments and your agreement that this work addresses a hard problem of causal inference in biological systems, and that the methodologies, datasets, metrics, and genetic assays are explained well. We address each of your concerns below.
> (AR: Author Response)
>
> **The title is misleading. The paper is much more about gene knockdown than drug discovery. As presented, causal connections between specific loss-of-function mutations and immunologic phenotypes may be drawn, but drug discovery is not directly related.**
>
> AR: Drug discovery is a complex pipeline that involves several sub-problems, from target identification to compound screening, safety profiling, and many others (see for example [1] below). While several benchmarks already exist for some of these sub-problems (see benchmarks listed in Section 2, in particular [2] focusing on de novo compound screening), there is a clear gap with respect to experimental design for target identification, which is what our benchmark focuses on. In that respect, we believe that our work is at the heart of drug discovery, but we have clarified further in the “Benchmarks” paragraph of Section 2 the differences in focus with existing benchmarks that are also falling under the “drug discovery” umbrella.
>
> [1] Hughes, J. P., Rees, S., Kalindjian, S. B., & Philpott, K. L. (2011). Principles of early drug discovery. British journal of pharmacology, 162(6), 1239–1249. https://doi.org/10.1111/j.1476-5381.2010.01127.
> [2] GuacaMol: Benchmarking Models for de Novo Molecular Design
> Nathan Brown, Marco Fiscato, Marwin H.S. Segler, and Alain C. Vaucher
> Journal of Chemical Information and Modeling 2019 59 (3), 1096-1108
> DOI: 10.1021/acs.jcim.8b00839
>
> **The authors do not provide any novel methodologies within the field of active learning or drug development.**
>
> AR: Please see our message to the AC on that point (“Guidance on how to review benchmark papers”): since this work is a benchmark paper, it has different objectives compared with other methodology-focused papers and is based on the ICLR reviewer guidelines (point 2.1), it should be assessed as such. In particular, prior benchmark papers accepted at ICLR in the past 3 years did not introduce new methodology in the areas they were focused on. We believe novelty for this type of work should be assessed with respect to the prior existence of other benchmarks in the area, and to our knowledge, there is no such benchmark, neither for biological experiment design nor more broadly for active learning in general. Consequently, we believe our work addresses an important gap that will be useful for several machine learning sub-communities.
>
> **The assays are limited to a small subset of immunologic phenotypes.**
>
> AR: Given your feedback and that of other reviewers, we will include two additional non-immunological assays [1, 2] to our benchmark in the camera-ready to cover a wider range of phenotypes (viral host factors and neuron tau modulation). We are currently running the experiments but, unfortunately, due to the computationally intensive nature of this work, it is likely that the experiments would not conclude before the revision deadline. Nonetheless, we are committed to adding them to the camera-ready.
>
> [1] Zhu et al. A genome-wide CRISPR screen identifies host factors that regulate SARS-CoV-2 entry. Nat Comms 2021
> https://www.nature.com/articles/s41467-021-21213-4
>
> [2] Sanchez et al. Genome-wide CRISPR screen identifies protein pathways modulating tau protein levels in neurons. Communications Biology 2021 https://www.nature.com/articles/s42003-021-02272-1
>
> --

---

### Decision · Program_Chairs · 2022-01-20

**Decision:**

Accept (Poster)

**Comment:**

This paper introduces a benchmark for experimental design algorithms
for an important cellular biological question, causal discovery of
effective genetic knock-out interventions. It uses existing datasets.

The paper was discussed by the reviewers after the authors correctly
pointed out that methodological machine learning novelty is not a
necessary condition for accepting papers. Two reviewers increased
their scores and all are slightly positive. The benchmark was seen as
valuable, and one reviewer even commented they might use it in their
own research. However, the paper is still on the borderline as this
benchmark is only a first step. It has not been shown yet that machine
learning insights can be produced with it, as the authors have not
actually used it for benchmarking yet. In other words, the benchmark
can be considered a potentially excellent idea which has not been
tested empirically yet.

This seems a highly promising research direction and the authors are
strongly encouraged to continue to providing the benchmarks and
releasing the method to the community so that others can help them in that.